# Background compensation revisited: Conserved phase response curves in frequency controlled homeostats with coherent feedback

Peter Ruoff *

Department of Chemistry, Bioscience, and Environmental Engineering, University of Stavanger, Stavanger, Norway

* ruoff@ux.uis.no

**Data Availability Statement:** All relevant data are within the article and its Supporting information files.

## Abstract

Background compensation is the ability of a controlled variable to respond to an applied perturbation in an unchanged manner and independent of different but constant background signals which act in parallel to the perturbation. Background compensation occurs by 'coherent feedback' mechanisms where additional control variables feed directly back to the controlled variable. This paper extends a previous study on background compensation to include phase responses in frequency controlled coherent feedback oscillators. While the frequency resetting amplitude in coherent feedback oscillators is found to be dependent on the inflow/outflow perturbation of the controlled variable and thereby become phase dependent, the frequency resetting itself and the corresponding phase response curves are found to be background compensated. It is speculated that this type of background compensation may be an additional way how ambient noise can be 'ignored' by organisms.

## Introduction

Homeostatic mechanisms play an essential role in the defense of organisms against environmental and internal disturbances and thereby contribute to their stability. The term 'homeostasis' was introduced in 1929 by Walter Cannon [1, 2] similar to Claude Bernard's concept [3] of the constancy of the internal milieu [4]. With the rise of cybernetics, Norbert Wiener related homeostasis to negative feedback mechanisms as 'exemplified in mechanical automata' [5]. In the following years Wiener's negative feedback concept of homeostasis was applied to various physiological examples [6]. However, by the end of the 1980's researchers became more critical that homeostasis would only relate to single negative feedback loops with a fixed setpoint. Focussing on properties such as variable setpoints, multiple feedbacks, the application to circadian rhythms, or on nonlinear dynamic behaviors alternative terms like 'predictive homeostasis' [7], 'allostasis' [8, 9], 'rheostasis' [10] or 'homeodynamics' [11, 12] were suggested instead and in addition [13] to 'homeostasis'. As pointed out by Carpenter [14], although homeostatic

**Funding:** The author(s) received no specific funding for this work.

**Competing interests:** The authors have declared that no competing interests exist.

regulation shows many different facets the term 'homeostasis' still can serve as an overarching concept.

In this paper I revisit theoretical work on oscillatory homeostats with robust frequency control. Frequency control has been observed in many biological systems, for example in circadian rhythms [15] or neuronal oscillations [16, 17]. While our understanding how frequency control is achieved in these systems is still relatively poor, I feel that insights into possible mechanisms of robust frequency and response control may be helpful to uncover principles behind such regulations.

## Motivation and aim of this work

The motivation of this work originates from a recent finding [18] that a certain type of negative feedback, termed coherent feedback, is able to show identical response kinetics upon a perturbing signal irrespective of different but constant backgrounds which act on the same reaction channel as the perturbation. This type of background compensation may be related to the ability of animals living in large colonies (for example bats, penguins, or seabirds) to filter out individual calls between partners or between a chick and a parent despite of a highly noisy environment.

However, there is still little knowledge on how a coherent feedback responds upon different inflow or outflow perturbations in an oscillatory context. The aim of this paper is to extend the earlier findings by including both inflow and outflow perturbations at the controlled variable. The applied inflow and outflow perturbations to and from *A* result in distinct phase dependencies in the oscillators' frequency resetting. However, despite these phase dependencies, the frequency resetting and the corresponding phase response curves (PRCs; for definition see below) turn out to be background compensated, while in ordinary single-loop feedback oscillators background compensation of the PRCs is not observed.

## Materials and methods

### Computational methods

Computations were performed with the Fortran subroutine LSODE [19] (https://computing.llnl.gov/projects/odepack). Plots were generated with gnuplot (www.gnuplot.info) and movies were made from a sequence of plots using QuickTime (https://support.apple.com/en-us/docs/software). Reaction schemes and plot annotations were prepared with Adobe Illustrator (www.adobe.com). Concentrations of compounds such as *A*, *E*, $I_1$,... are described by their compound names without square brackets. Rate constants and other parameters are in arbitrary units (au) and represented by $k_1$, $k_2$, $k_3$,... independent of their kinetic meaning, i.e. whether they are turnover numbers, Michaelis constants, or inhibition constants.

For documentation a set of selected computations are made available as Python and Matlab scripts (see S1 Program).

### Calculating averages

Averages of an oscillatory compound *X* are described as $<X>$ and have been calculated by two methods. In one method *X* is integrated for a given time period $\tau$ and the average is determined as the ratio between the integral of *X* and $\tau$, i.e.:

$$< X > (\tau) = \frac{1}{\tau} \int_0^\tau X(t)\, dt \qquad (1)$$

However, this method depends on the history of integrated *X*, which sometimes, especially

after a rapid change in $X$, has the disadvantage that $<X>(\tau)$ may only slowly converge to its true value at $\tau$.

In the other method I have used a self-chosen number ($N_{sw}$) of overlapping time intervals or 'sliding windows' with time length $\Delta t$, i.e. $[t_i, t_i + \Delta t]$. The averages of $X$ within each single window $i$ is

$$<X>_i = \frac{1}{\Delta t} \int_{t_i}^{t_i+\Delta t} X(t)\, dt \qquad (2)$$

The $t_i$'s are equal to LSODE's step length and represent the successive time points during LSODE's numerical integration. For this 'sliding window method' $<X>$ is calculated as:

$$<X> = \frac{1}{N_{sw}} \sum_{j=1}^{N_{sw}} <X>_j \qquad (3)$$

As a convention, $<X>$ is placed in the $X$-time plot at the time-middle of the last sliding window when $j = N_{sw}$.

## Negative feedback structures used in this study

Drengstig et al. suggested a set of eight two-component ($A$ and $E$) negative feedback loops termed Motif 1 (M1) up to Motif 8 (M8), where $A$ is the controlled variable and $E$ represents the manipulated/controller variable (see Fig 1 in Ref [20]). Depending whether the $E$-originating compensatory fluxes add or remove $A$ the eight motifs divide into two equal classes which have been termed 'inflow' or 'outflow' controllers, respectively. Several conditions have been described in the literature to obtain integral control and robust perfect adaptation [21–26], where zero-order [20, 27–31], first-order [32–34], or second-order (antithetic) [35–38] kinetics play essential roles.

When using zero-order removal of both $A$ and $E$, oscillatory behavior in the eight controller motifs has been observed [39]. In this study M2 and M8 will be used as oscillatory controllers. The M2 motif is an inflow controller closely related to Goodwin's 1963 oscillator [40, 41] and exhibits homeostasis in $<A>$. M8, on the other hand, is an outflow controller and in its oscillatory mode the M8 feedback shows homeostasis in the $A$-inhibited flux which is directed to $E$.

## Usage of zero-order kinetics

The M2 and M8 schemes I use include zero-order kinetics for two reasons: firstly, to introduce robust perfect adaptation in the controlled variables (see below), and secondly, to promote oscillations. Concerning the promotion of oscillations, Goodwin [40] presented in 1963 a two-variable negative feedback oscillator which since has been the basis for many physiological model oscillators [41]. An essential aspect in Goodwin's 1963 oscillator is the presence of zero-order degradation of the two components. Thorsen et al. [39] showed that for conservative two-component negative feedback oscillators the above zero-order assumption leads to a general equation of the form

$$\frac{\ddot{X}}{f^2} + X = X_{ss} \qquad (4)$$

where $X$ can be either of the two feedback variables, $X_{ss}$ is the steady-state expression of $X$, where $f$ approximately describes the frequency of the conservative oscillator (for details see Supporting Information in Ref [39]). In this respect, the conservative oscillator schemes which arise by the zero-order conditions can be viewed as a driving force for oscillations even when

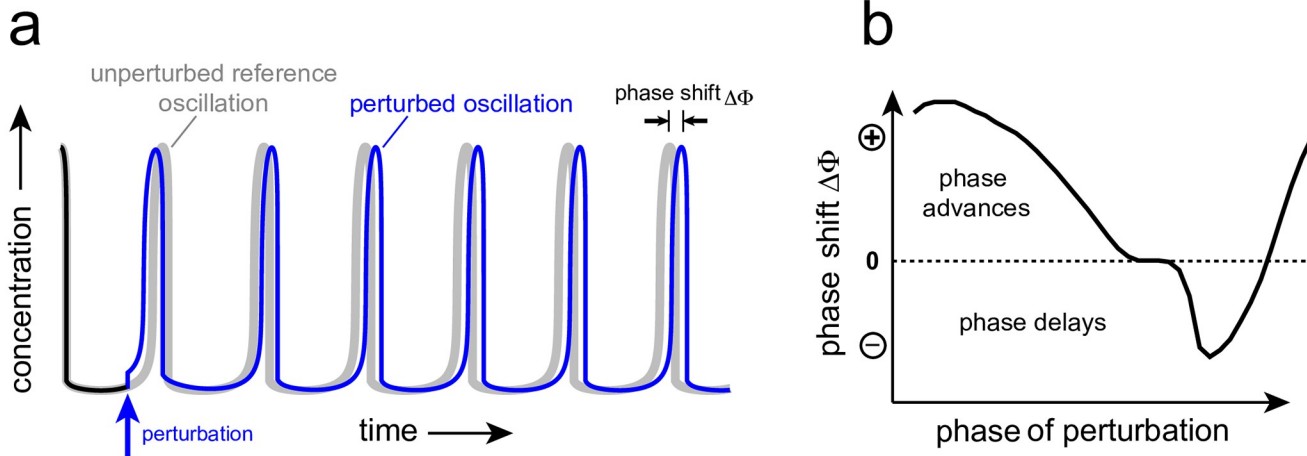

**Fig 1. Determination of a phase response curve.** Panel a: The application of a perturbation (indicated by the blue arrow) causes a phase shift ΔΦ between corresponding maxima of the perturbed and unperturbed oscillations. Panel b: A phase response curve is constructed by plotting phase shifts against the phases of perturbations, which are applied within one cycle of the unperturbed oscillation.

intermediates are present within the feedback loop. Results along similar lines were also found by Kurosawa and Iwasa [42], who observed that introducing Michaelis-Menten kinetics in the degradation reactions of circadian clock models promote oscillations.

## Phase response curves

We have used phase response curves to study the resetting dynamics of step- and pulse-perturbed oscillators. Phase responses have extensively been used in biology, especially in the study of circadian rhythms [43–46], but also in mechanistic analyses of purely chemical oscillators [47–49]. Fig 1 illustrates the method to determine a phase response curve. A perturbation (step or pulse) is applied at a certain phase and the resulting train of oscillations, outlined in blue, is compared with the corresponding undisturbed oscillator, which is outlined in gray (panel a). In a phase response curve (Fig 1b) the phase difference or phase shift ΔΦ between corresponding peaks of perturbed and unperturbed oscillations is plotted against the phase of perturbation.

In biology positive phase shifts are generally related to *phase advances*, while negative phase shifts relate to *phase delays*, which leads to the definition of ΔΦ as

$$\Delta\Phi = t^{max}_{reference} - t^{max}_{perturbed} \tag{5}$$

where $t^{max}_{reference}$ is the time of a maximum of the unperturbed oscillation, while $t^{max}_{perturbed}$ is the time of the corresponding maximum after the perturbation has been applied.

## Structure of the paper

The 'Results and discussion' section is divided into two parts. The first part ('Motif 2 based controllers') covers the behaviors of M2 based negative feedback oscillators, while the second part ('Motif 8 based controllers') deals with M8 oscillators. M2 and M8 motifs behave slightly different as M2 feedbacks show homeostasis in the concentration of the controlled variable termed *A*, while oscillatory M8 feedbacks show homeostasis of the *A*-inhibited flux which is

directed to the manipulated variable $E$. Each of the M2 and M8 related parts are structured as follows:

- First I deal with single-loop oscillations, describe their homeostatic properties and their operational limits due to the inflow/outflow properties of the respective compensatory fluxes.

- Then coherent feedbacks are introduced by the inclusion of two additional controllers $I_1$ and $I_2$, which feed directly back to $A$. The inclusion of $I_1$ and $I_2$ extends the single-loop operational limits and rescues the oscillatory controller from breakdown. In addition, frequency homeostasis as well as background compensation in frequency resetting are now observed.

- Finally the PRCs between single-loop and coherent feedbacks are compared. Coherent feedback shows the ability to compensate phase shifts against different backgrounds and thereby leaves PRCs unaltered.

The paper ends with a brief summary and an outlook for further research.

## Results and discussion

### Motif 2 based controllers

**M2 single-loop: Integral control of $A$ concentration oscillations and controller breakdown by a dominant inflow to $A$.**   Fig 2 shows a single-loop M2 feedback scheme which is related to Goodwin's 1963 oscillator.

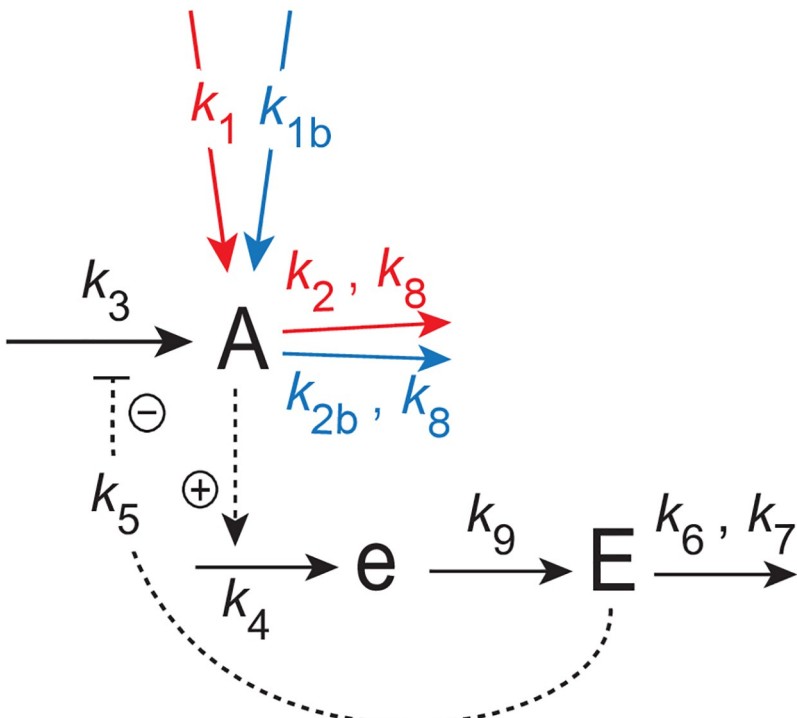

**Fig 2. Feedback arrangement of a single-loop feedback similar to Goodwin's 1963 oscillator, which is based on motif 2 [20].** Compound $A$ is the controlled variable, while $E$ is the controller. Compound $e$ is an intermediate which assures limit cycle oscillations when the system is oscillatory. Red reaction arrows indicate applied step perturbations. The blue arrows indicate constant backgrounds.

The rate equations are:

$$\dot{A} = \underbrace{\frac{k_3 \cdot k_5}{k_5 + E}}_{\text{compensatory flux } j_3} - \underbrace{\frac{k_2 \cdot A}{k_8 + A}}_{\text{perturbation}} - \underbrace{\frac{k_{2\text{b}} \cdot A}{k_8 + A}}_{\text{background}} + \underbrace{k_1}_{\text{perturbation}} + \underbrace{k_{1\text{b}}}_{\text{background}} \tag{6}$$

$$\dot{e} = k_4 \cdot A - k_9 \cdot e \tag{7}$$

$$\dot{E} = k_9 \cdot e - \frac{k_6 \cdot E}{k_7 + E} \tag{8}$$

Dependent on the perturbation on $A$ variable $E$ controls $A$ by acting either as a repressor or derepressor thereby adjusting the compensatory flux $j_3 = k_3 k_5/(k_5 + E)$. To obtain integral control in variable $A$ (or $<A>$ when the system is oscillatory) the difference between the setpoint of $A$ or $<A>$ and its actual value is calculated, which is integrated in time. The integrated error is then used to correct for an applied perturbation in $A$ [21–24]. The approach taken here uses zero-order kinetics with the interpretation [20, 29] that certain control-related $E$-removing enzymes work under saturated or near saturation condition. Enzymes showing zero-order kinetics bind strongly to their substrates and have a low Michaelis constant ($K_M$) compared with the enzyme's substrate concentration [50]. In the calculations the zero-order condition is normally assured when using a $K_M$ (here $k_7$) in the order between $10^{-4}$ and $10^{-6}$ au. Higher $K_M$ values will lead to a diminished homeostatic accuracy of the controller (see for example S9 Fig in [20]).

Since the scheme in Fig 2 is forced to be oscillatory by zero-order removals of $A$ and $E$ we assume that the average concentrations in $A$, $E$, and $e$ are at steady state, i.e. $<\dot{A}> = <\dot{e}> = <\dot{E}> = 0$. The setpoint of the controlled $<A>$ variable is $k_6/k_4$, which is obtained by combining Eqs 7 and 8 and eliminating the term $k_9 \cdot e$. Taking then the average and solving for $<A>$ we get:

$$k_4 \cdot <A> - k_6 \cdot \underbrace{\left\langle \frac{E_{ss}}{k_7 + E_{ss}} \right\rangle}_{\approx 1} = 0 \quad \Rightarrow \quad <A> = A_{set} = \left( \frac{k_6}{k_4} \right) \tag{9}$$

The feedback in Fig 2 is an inflow controller [20], which points to the fact that the compensatory flux $j_3 = k_3 k_5/(k_5 + E)$ is an inflow to the controlled variable $A$ and thereby compensating outflows from $A$. The range of $k_2$ for which the feedback can show homeostatic control in $<A>$ is limited by the following two conditions:

(i) by the maximum average compensatory flux $<j_3> = k_3$, which is reached when $k_2 \to k_2^{\max}$ and $E \to 0$ with

$$k_2^{\max} = k_3 + (k_1 + k_{1\text{b}}) - k_{2\text{b}} \tag{10}$$

from Eq 6. Controller breakdown occurs when $k_2 \geq k_2^{\max}$.

(ii) by the minimum compensatory flux $<j_3> = 0$, which is reached when the total inflow to $A$ balances or becomes larger than the total outflow from $A$, i.e.

$$(k_1 + k_{1\text{b}}) \geq k_2 + k_{2\text{b}} \tag{11}$$

In this case $A$ and $E$ increase spontaneously (termed windup) while the system tries to satisfy the formal relationship $j_3 = k_3 k_5/(k_3 + E) = 0$. Fig 3 illustrates this type of controller breakdown by a successive increase of $k_1$. As we will show below the limitation due to Eq 11

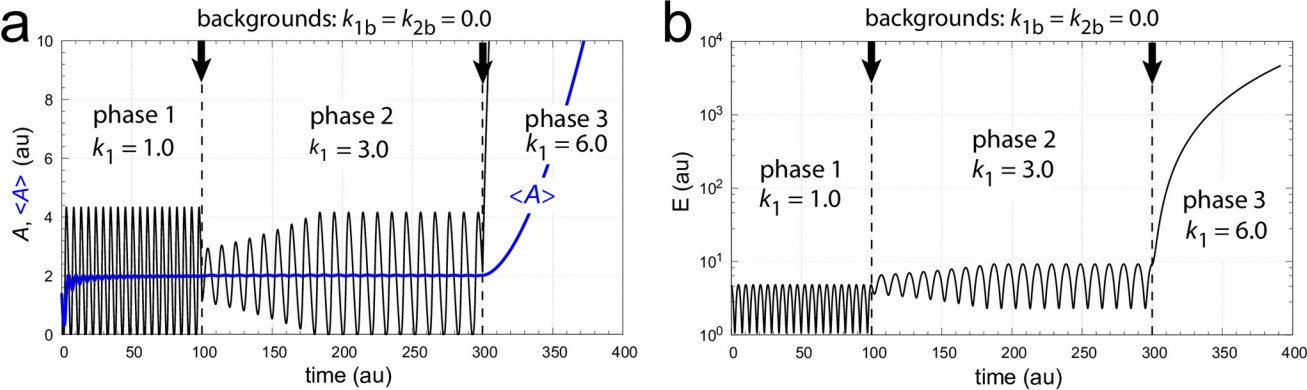

**Fig 3. Illustration of controller breakdown of the scheme in Fig 2 when inflows to *A* exceed the outflows from *A*.** The setpoint of $<A>$ is 2.0. Panel a: *A* (in black) and $<A>$ (in blue) are shown as a function of time when $k_1$ increases successively from $k_1$=1.0 (phase 1) to $k_1$=3.0 (phase 2), and finally to $k_1$=6.0 in phase 3. The final $k_1$ satisfies Eq 11 which leads to controller breakdown and to a rapid increase in *A*. The average $<A>$ is calculated after Eq 1. Vertical arrows show the times the $k_1$ steps are increased. Also note the decrease in the oscillator's frequency when $k_1$ gets larger. Panel b: *E* as a function of time for the same $k_1$ steps as in panel a. Frequency phase 1: 0.190; frequency phase 2: 0.097. Rate constants: $k_{1b}$=0.0, $k_2$=5.0, $k_{2b}$=0.0, $k_3$=100.0, $k_4$=1.0, $k_5$=0.1, $k_6$=2.0, $k_7$=$k_8$=1×10$^{-6}$, $k_9$=20.0. Initial concentrations: $A_0$=1.407, $E_0$=4.754, $e_0$=7.524×10$^{-2}$.

can be circumvented in a frequency-controlled version of the single-loop feedback when 'outer control loop' species $I_1$ and $I_2$ act as inflow and outflow controllers to both *A* and *E*.

There is also a third condition of controller breakdown when one of the signaling events (indicated by the dashed lines in Fig 2) reach saturation [31]. For the sake of simplicity we have not considered this possibility here and assumed that the activation of *e* by *A* is described by first-order kinetics with respect to the activator, i.e. $j_4 = k_4 \cdot A$ without an activation constant.

**M2 coherent feedback: Frequency control and background compensation in frequency resetting.** An interesting feedback arrangement occurs when additional controllers $I_1$ and $I_2$ in Fig 4 feed directly back to *A*. In this case robust homeostasis in both *A* and *E* can be achieved. The control of *E* by $I_1$ and $I_2$ leads in addition to frequency homeostasis [39]. In analogy to a similar feedback arrangement in quantum control theory [51, 52] we have termed the feedback scheme in Fig 4 as 'coherent feedback' [18].

The rate equations are:

$$\dot{A} = \underbrace{k_1}_{\text{perturbation}} + \underbrace{k_{1b}}_{\text{background}} + k_{g3} \cdot I_2 + \frac{k_3 \cdot k_5}{k_5 + E} - \frac{k_g \cdot A \cdot I_1}{k_{17} + A} - \underbrace{\frac{k_2 \cdot A}{k_8 + A}}_{\text{perturbation}} - \underbrace{\frac{k_{2b} \cdot A}{k_8 + A}}_{\text{background}} \tag{12}$$

$$\dot{e} = k_4 \cdot A - k_9 \cdot e \tag{13}$$

$$\dot{E} = k_9 \cdot e - \frac{k_6 \cdot E}{k_7 + E} \tag{14}$$

$$\dot{I}_1 = k_{11} \cdot E - \frac{k_{12} \cdot I_1}{k_{13} + I_1} \tag{15}$$

$$\dot{I}_2 = k_{14} - \left(\frac{k_{15} \cdot I_2}{k_{16} + I_2}\right) \cdot E \tag{16}$$

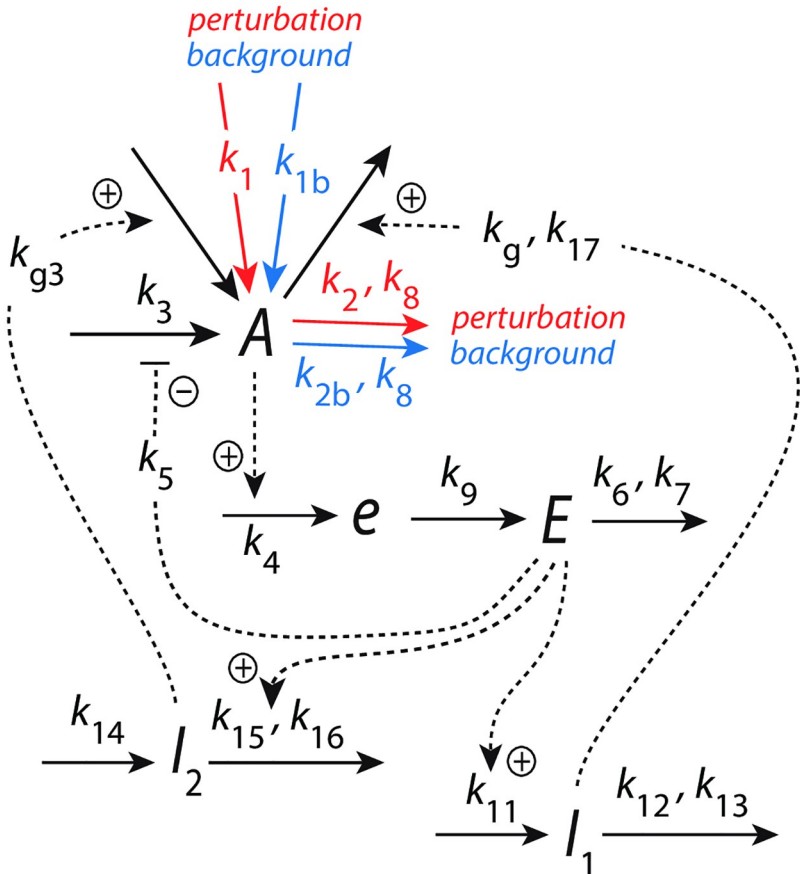

**Fig 4. Coherent feedback keeps $<A>$, $<E>$ and the frequency under homeostatic control by $I_1$ and $I_2$, which directly feed back to variable $A$.**

$k_1$ or $k_2$ denote step perturbations while $k_{1b}$ and $k_{2b}$ represent constant backgrounds. The setpoint of $<A>$ is still given by Eq 9, while two setpoints for $<E>$ are obtained from the rate equations of respectively $I_1$ and $I_2$, i.e. [18, 39]

$$E_{set}^{I_1} = \frac{k_{12}}{k_{11}}; \qquad E_{set}^{I_2} = \frac{k_{14}}{k_{15}} \tag{17}$$

For the sake of simplicity we here keep both $E_{set}^{I_1}$ and $E_{set}^{I_2}$ rather arbitrary equal to 5.0 and $A_{set}$ to 2.0.

The interesting aspect of the coherent feedback scheme in Fig 4 is the fact that it can compensate for different but constant background perturbations. In our previous work [18] only *outflow* perturbations were considered, because the central feedback loop *A-e-E-A* in Fig 4 (being an *inflow* controller) compensates essentially for *outflow* perturbations in *A* [20]. However, since the $I_1$-$I_2$ 'outer feedback layer' in Fig 4 should also allow to compensate for inflows to *A*, as implied by the work of Thorsen [53], I tested the coherent feedback scheme with respect to inflow perturbations to *A*. In Fig 5 we have the same rate constants and perturbative conditions as in Fig 3, but keep both *A* and *E* under homeostatic control by $I_1$ and $I_2$ (see Fig 5a and 5b). Fig 5c shows that the increasing $k_1$ steps lead to an increase of $I_1$ and a decrease of

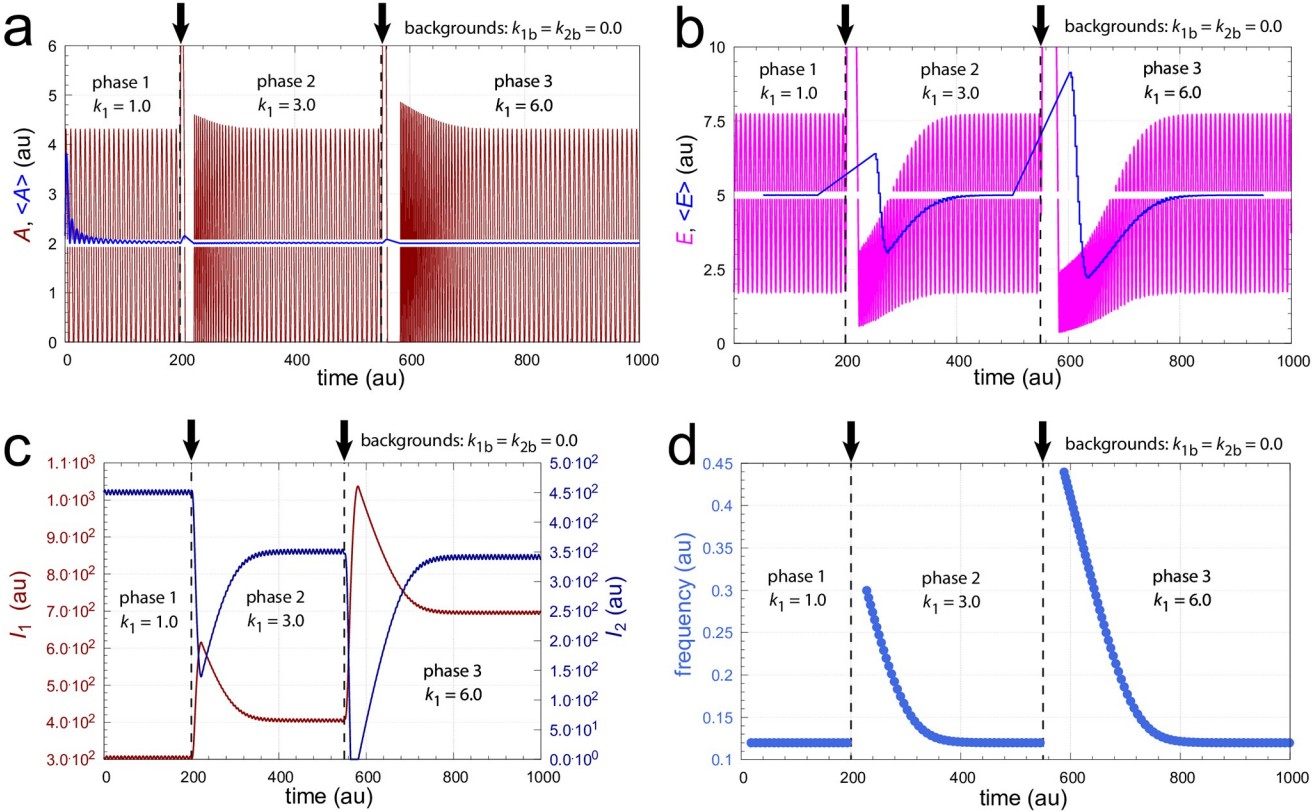

**Fig 5. Compensation of inflow perturbations to $A$ by controllers $I_1$ and $I_2$.** Panel a: $A$ and $<A>$ are shown as a function of time. The setpoint of $A$ is 2.0. Blue line shows $<A>$ calculated by Eq 1. Panel b: $E$ and $<E>$ as a function of time. The setpoint of $E$ is 5.0. Blue line shows $<E>$ calculated by Eqs 2 and 3 with $\Delta t$ (sliding window size)=100.0 time units, $N_{sw}$=50, and step length = 0.05. Panel c: $I_1$ and $I_2$ as a function of time. Panel d: Frequency as a function of time. Vertical arrows indicate the time points when the $k_1$ steps occur. Rate constants: $k_{1b}$=0.0, $k_2$=5.0, $k_{2b}$=0.0, $k_3$=100.0, $k_4$=1.0, $k_5$=0.1, $k_6$=2.0, $k_7$=$k_8$=1×10$^{-6}$, $k_9$=20.0, $k_{11}$=1.0, $k_{12}$=5.0, $k_{13}$=1×10$^{-6}$, $k_{14}$=5.0, $k_{15}$=1.0, $k_{16}$=$k_{17}$=1×10$^{-6}$, $k_g$=$k_{g3}$=0.01. Initial concentrations: $A_0$=2.080, $E_0$=1.731, $e_0$=9.677×10$^{-2}$, $I_{1,0}$=304.87, $I_{2,0}$=450.57.

$I_2$ which both contribute to the compensation of the inflow to $A$. Fig 5d shows that the steady state frequency is independent of $k_1$.

Since the scheme in Fig 4 is stable against inflows and outflows to and from $A$ I have tested its frequency resetting behavior for step perturbations by both $k_1$ and $k_2$. Fig 6 shows surprising differences in the frequency resetting: when $A$ is perturbed by $k_1$ steps the frequency resetting is highly dependent on the phase of the perturbation (Fig 6a). On the other hand, when $k_2$ outflow steps are applied, the resetting of the frequency is practically independent of the phase where the perturbation is applied (Fig 6b).

A closer look reveals that the difference in the two resetting behaviors is caused by the topology of the phase space kinetics. When $k_1$ steps are applied the trajectory makes large excursions in phase space and returns to its oscillatory state after a transient. In case of $k_2$ steps the system is rapidly pushed into an oscillatory state via a very short transit at low $A$ concentrations along the $I_1$-$E$ or $I_2$-$E$ manifolds. Figs 7 and 8 show the time profiles of $A$, $E$, $I_1$, $I_2$, and the frequency when $k_1$ and $k_2$ steps are respectively applied at time $t$ = 100. In both Figs 7 and 8 the setpoints of $A_{set}$=2.0 and $E_{set}$=5.0 are defended due to the compensatory actions by $I_1$ and $I_2$, which results in the frequency homeostasis of the oscillator.

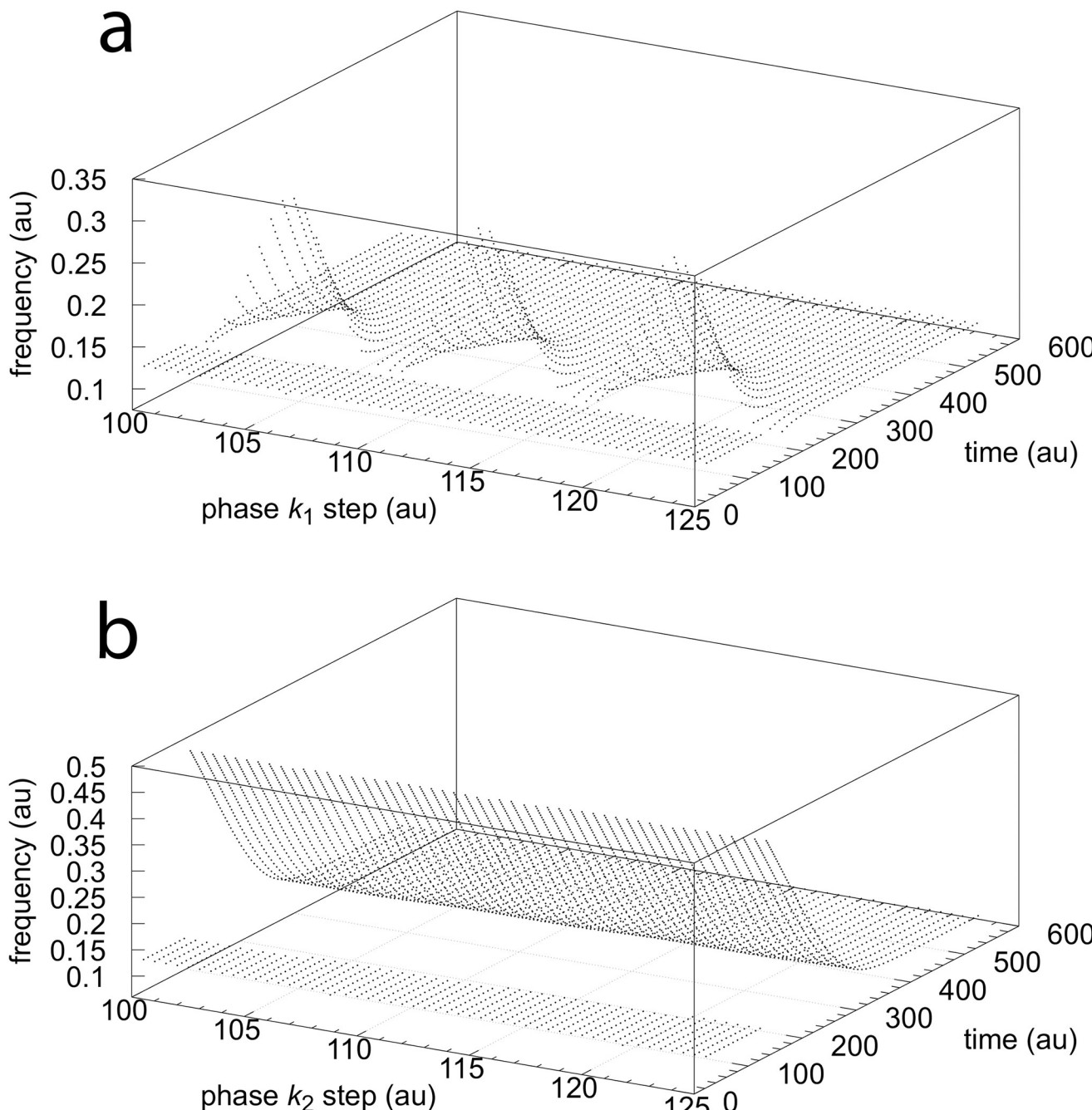

**Fig 6. Phase dependencies of the frequency resetting in the oscillator Fig 4.** Panel a: a $k_1$ step perturbation of $1.0{\rightarrow}3.0$ is applied at different times/phases starting at t = 100.0 and ending at t = 125.0 with intervals of 0.5. At each phase the resetting frequency is plotted against time. Rate constants: $k_{1b}$=20.0, $k_2$=1.0, $k_{2b}$=32.0, $k_3$=100.0, $k_4$=1.0, $k_5$=0.1, $k_6$=2.0, $k_7$=$k_8$=1×10$^{-6}$, $k_9$=20.0, $k_{11}$=1.0, $k_{12}$=5.0, $k_{13}$=1×10$^{-6}$, $k_{14}$=5.0, $k_{15}$=1.0, $k_{16}$=$k_{17}$=1×10$^{-6}$, $k_g$=$k_{g3}$=0.01. Initial concentrations: $A_0$=1.262, $E_0$=7.550, $e_0$=6.625×10$^{-2}$, $I_{1,0}$=2766.7, $I_{2,0}$=3709.6. Panel b: As panel a, but a $k_2$ step perturbation of $1{\rightarrow}10$ is applied with $k_1$=1.0. Other rate constants, backgrounds $k_{1b}$ and $k_{2b}$, and initial concentrations as in panel a.

Fig 9 shows the trajectories of Figs 7 and 8 in $A\text{-}E\text{-}I_1$ and $A\text{-}E\text{-}I_2$ phase space. In case of the $k_1$ step (Fig 9a and 9b) the trajectory makes a relative large excursion before settling to the limit cycle after the step. This large excursion is the cause for the observed phase dependency in the frequency resetting seen in Fig 6a. For the $k_2$ step, however, the excursion of the

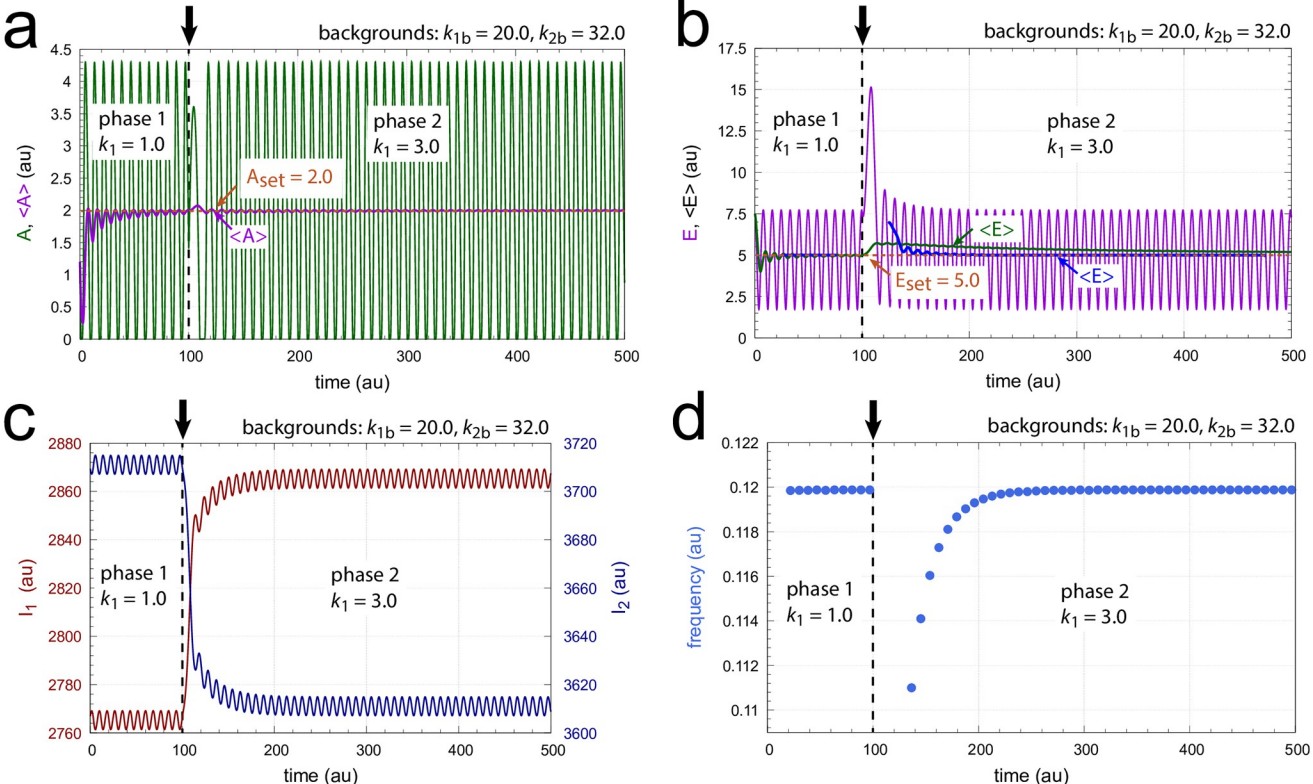

**Fig 7. Time profiles of A, E, $I_1$, $I_2$, and the frequency when a $k_1$ : 1.0 → 3.0 step is applied at time $t$ = 100.0 (indicated by the vertical arrows).** Panel a: Concentration of A as a function of time. Averages <A> are calculated by Eq 1. Panel b: Concentration of E as a function of time. Averages <E> (green line) are calculated by Eq 1 while <E> values outlined in blue are calculated by Eqs 2 and 3 using 500 time intervals of 50 units in phase 1 and 3500 intervals of the same extension in phase 2. Panel c: Concentrations of $I_1$ and $I_2$ as a function of time. Panel d: Frequency as a function of time. Rate constants and initial concentrations as in Fig 6.

trajectory is minor and moves quickly into the high frequency regime of the phase space leading to the frequency resetting as shown in Fig 6b. The files in S1 Movie show the moving trajectories for the $k_1/k_2$ perturbations and the preservation of the projected limit cycles on to the A-E phase space.

While Fig 9 and S1 Movie provide insights into why the resetting of $k_1$ steps is markedly different from those of $k_2$ steps, in the next calculations I tested whether background compensation is operative for both $k_1$ and $k_2$ step perturbations. This is shown in Figs 10 and 11, where two different backgrounds, ($k_{1b}$=20.0 & $k_{1b}$=32.0, large orange dots) and ($k_{1b}$=20.0 & $k_{1b}$=40.0, small blue dots), are applied. Background compensation is indicated by the fact that both backgrounds show identical frequency resettings, since orange and blue dots lie precisely at the same location.

S2 and S3 Movies show Figs 10a and 11a with varying viewing angles.

**M2 oscillators: Background influences on PRCs.** Phase response curves (PRCs) are an often used tool to analyze biological or chemical oscillators. Especially have PRCs been used in relationship with circadian rhythms [43–46]. Due to the different phase resettings when $k_1$ steps are applied to the oscillator in Fig 4 (see Figs 6a and 10a) I became interested to what extend phase shifts may be influenced by the applied backgrounds $k_{1b}$ and $k_{2b}$. Since increased backgrounds can lead to a diminished response amplitude in analogy to Weber's law [54, 55],

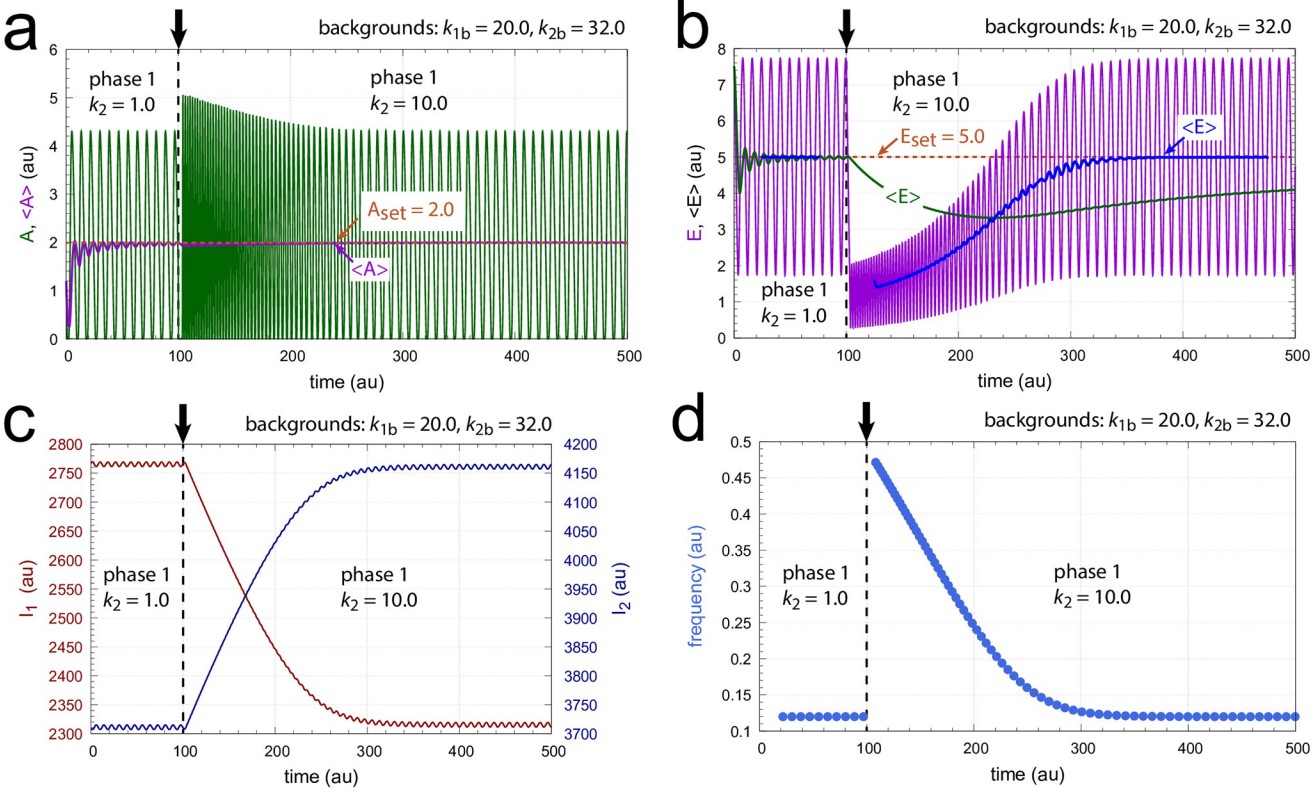

**Fig 8. Time profiles of *A*, *E*, *I*₁, *I*₂, and the frequency when a *k*₂ : 1.0 → 10.0 step is applied at time *t* = 100.0 (indicated by the vertical arrows) with *k*₁=1.0.** Averages <*A*> and <*E*> (green line) are calculated by Eq 1 while <*E*> values outlined in blue are calculated by Eqs 2 and 3 as in Fig 7. Other rate constants and initial concentrations as in Fig 6.

one may expect that Weber's law may also apply to phase shifts. This expectation is, however, only partially fulfilled.

**PRCs of single-loop M2 oscillator.** To investigate the influence of backgrounds on PRCs I first consider the single feedback loop in Fig 2 before turning to the coherent feedback scheme of Fig 4. A problem with the oscillator in Fig 2 is that the period will depend on the inflow/outflow rates to and from *A* (Fig 3) and thereby backgrounds have an influence on the oscillator's frequency. In addition, phase shifts very often reach their final values first after a couple of cycles, which is illustrated in Fig 12. Fig 12a shows the application of a *k*₂ pulse at phase *t* = 1.0 leading to phase advances (positive phase shifts), while in panel b the pulse is applied at *t* = 15.0 which leads to phase delays.

The PRCs are constructed by plotting the final phase shifts (see Fig 12) against the phase of perturbation. To directly compare the PRCs at different backgrounds the 'phase of perturbation' is normalized with respect to the oscillator's period length. Phase = 0 is defined to occur at an *A* maximum while the next maximum in *A* defines the normalized phase to be 1. Fig 13 shows the normalized phase response curves for different *k*₂ᵦ backgrounds when the same *k*₂ pulse as in Fig 12 is applied. One sees a clear background dependency of the phase response curves: both the phase shift amplitude and the length of the dead zone (characterized by ΔΦ = 0) increase with increasing *k*₂ᵦ.

Concerning the dead zone, Uriu and Tei [56] recently found that saturation kinetics in the repressor synthesis appears responsible for the occurrence of a dead zone in circadian

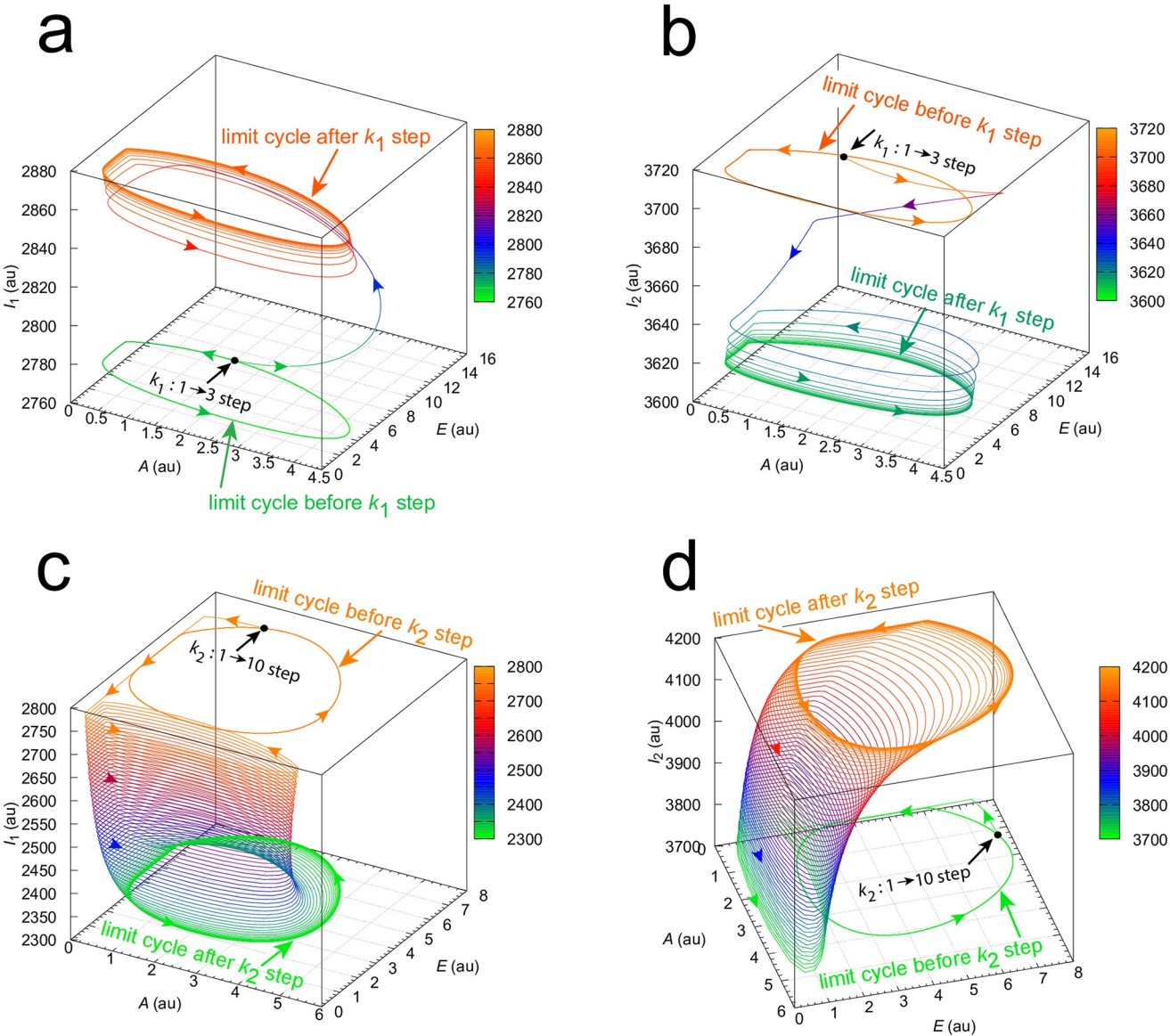

**Fig 9. $A$-$E$-$I_1$ and $A$-$E$-$I_2$ phase space trajectories of the perturbed oscillators in Figs 7 and 8.** Panels a and b show the trajectories for the $k_1 : 1.0 \rightarrow 3.0$ step at $t = 100$ in Fig 7. Panels c and d show the trajectories for the $k_2 : 1.0 \rightarrow 10.0$ step at $t = 100$ in Fig 8. Colors indicate the levels of $I_1$ or $I_2$.

rhythms. However, in our case the repressor kinetics nor the compensatory flux $j_3$ (see Eq 6) are saturated, but depend on the perturbations and backgrounds. What is saturated in our model are the removals of $A$ and $E$. This indicates that homeostatic constraints by zero-order kinetics may in addition lead to the appearance of a dead zone in the PRCs of circadian rhythms, an aspect which needs further investigations.

**PRCs of coherent M2 oscillator.** Next I applied $k_1 : 1.0 \rightarrow 128.0$ and $k_2 : 1.0 \rightarrow 128.0$ step perturbations on the oscillator with coherent feedback (Fig 4) using the rather arbitrary chosen three backgrounds: (i) $k_{1b}$=1.0 and $k_{1b}$=500.0, (ii) $k_{1b}$=1.0 and $k_{1b}$=10.0, and (iii) $k_{1b}$=100.0 and $k_{1b}$=10.0. To illustrate the procedure, Fig 14 shows the resetting and the determined phase shifts when the phase of perturbation is 3.0 (panel a) and 4.4 (panel c) with backgrounds

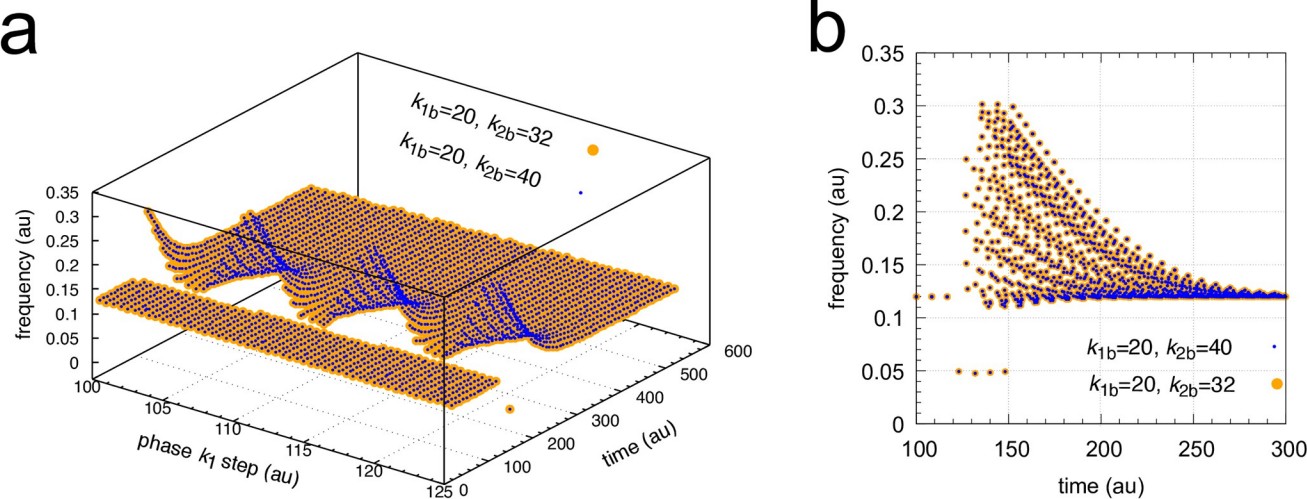

**Fig 10. Background compensation in frequency resetting for $k_1$ steps.** $k_1 : 1.0 \rightarrow 3.0$ steps at two backgrounds are applied. Orange dots: $k_{1b}$=20.0 and $k_{1b}$=32.0; blue dots: $k_{1b}$=20.0 and $k_{1b}$=40.0. Panel a shows frequency as a function of phase of applied $k_1$ steps and time, while panel b shows part of the projection on to the frequency-time axes. Rate constants as in Fig 6. Initial concentrations at $k_1$ phase = 0: $A_0$=4.304, $E_0$=3.952, $e_0$=2.150×10$^{-1}$, $I_{1,0}$=2761.6, $I_{2,0}$=3714.7. See also S2 Movie.

$k_{1b}$=1.0 and $k_{1b}$=500.0. Grayed oscillations in panels a and c represent the unperturbed rhythms, while the red oscillations show the effect of the perturbations. Panels b and d show the changes in phase shifts as a function of peak number and the final settling of the phase shift.

Fig 15 shows the calculated phase response curves of $k_1$ and $k_2$ $1.0 \rightarrow 128.0$ steps for the three backgrounds (i)-(iii) indicated above. The PRCs are completely congruent, although slight differences in the final phase shifts are observed when phase shifts are outside of the

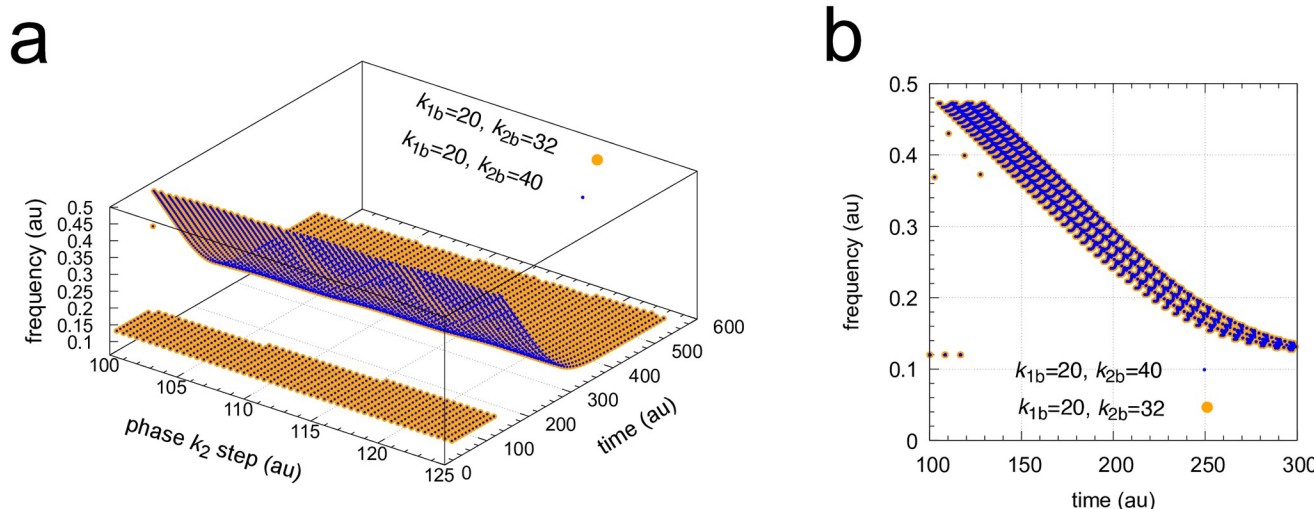

**Fig 11. Background compensation in frequency resetting for $k_2$ steps.** $k_2 : 1.0 \rightarrow 10.0$ steps at two backgrounds are applied. Orange dots: $k_{1b}$=20.0 and $k_{1b}$=32.0; blue dots: $k_{1b}$=20.0 and $k_{1b}$=40.0. Panel a shows frequency as a function of phase of applied $k_2$ steps and time, while panel b shows part of the projection on to the frequency-time axes. Rate constants as in Fig 6. Initial concentrations at $k_2$ phase = 0 as in Fig 10. See also S3 Movie.

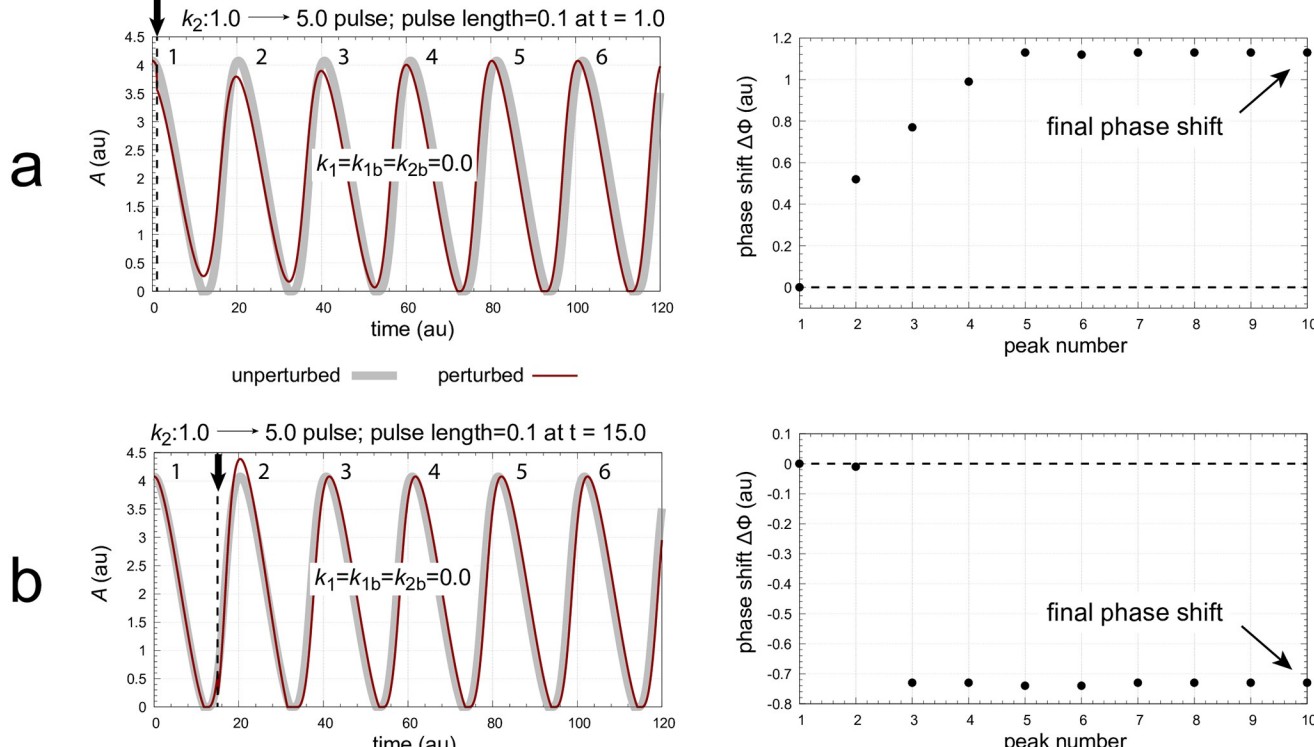

**Fig 12. Positive and negative phase shifts are observed when $k_2 : 1.0 \rightarrow 5.0$ pulses of 0.1 time units are applied to the oscillator of Fig 2.** Figures a and b, left panels: oscillatory concentrations of $A$ for the unperturbed and perturbed system (outlined in gray and red) when the pulse is applied at respectively 1.0 and 15.0 time units. Peak numbers are indicated near peaks. Figures a and b, right panels: phase shifts as a function of the peak numbers from left panels. The unperturbed oscillator has a period length of 20.342 time units. Rate constants: $k_1=k_{1b}=k_{2b}=0.0$, $k_3=100.0$, $k_4=1.0$, $k_5=0.1$, $k_6=2.0$, $k_7=k_8=1\times10^{-6}$, $k_9=20.0$. Initial concentrations: $A_0=4.0763$, $E_0=9.9288$, and $e_0=0.2038$.

constant phase shift zone. Surprisingly, this constant phase shift zone resembles that of a dead zone, but the final phase shift values are either negative (Fig 15a) or positive (Fig 15b).

## Motif 8 based controllers

In the next set of calculations I show results for the oscillatory M8 feedback scheme [20].

**M8 single-loop: Integral control of $A$-regulated flux and controller breakdown by a dominant outflow of $A$.**   Fig 16 shows the scheme of the considered M8 single negative feedback loop with $e$ as a precursor of controller $E$. The role of the addition of $e$ to the feedback is to turn an otherwise conservative system into a limit cycle oscillator.

This controller is also based on derepression by the manipulated variable $E$, but acts as an outflow controller, i.e. compensates for inflow perturbations to $A$. Unlike the M2 controller in Fig 2 the oscillatory M8 feedback does not control the concentration of $A$, but keeps the average of the $A$-regulated flux $<j_6> = <k_6 k_{10}/(k_{10}+A)>$ directed to $e$ constant. The rate equations are:

$$\dot{A} = \underbrace{k_1 - \frac{k_2 \cdot A}{k_8 + A}}_{\text{perturbations}} + \underbrace{k_{1b} - \frac{k_{2b} \cdot A}{k_8 + A}}_{\text{backgrounds}} - \underbrace{\left(\frac{k_4 \cdot A}{k_5 + A}\right) \cdot \left(\frac{k_9}{k_9 + E}\right)}_{\text{compensatory flux } j_4} \qquad (18)$$

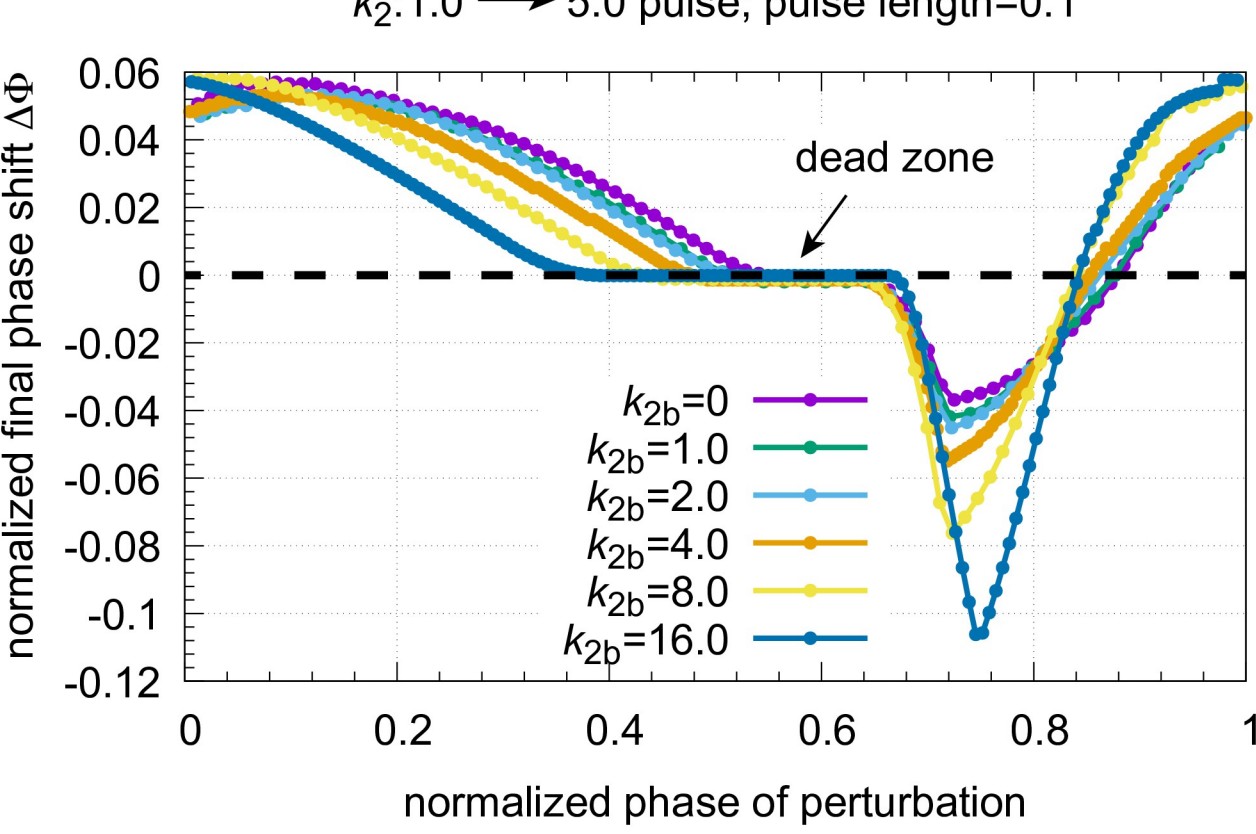

**Fig 13. Normalized phase response curves at different $k_{2b}$ backgrounds when a $k_2 : 1.0 \rightarrow 5.0$ pulse of 0.1 time length is applied to the oscillator of Fig 2.** Initial concentrations, $k_{2b}$=0.0: see legend Fig 12; initial concentrations $k_{2b}$=1.0: $A_0$=4.1570, $E_0$=4.9277, $e_0$=0.2078; period length: 10.265; initial concentrations $k_{2b}$=2.0: $A_0$=4.2416, $E_0$=3.2540, and $e_0$=0.2119, period length: 6.919; initial concentrations $k_{2b}$=4.0: $A_0$=4.4218, $E_0$=1.9125, and $e_0$=0.2381, period length: 4.263; initial concentrations $k_{2b}$=8.0: $A_0$=4.8168, $E_0$=1.0398, and $e_0$=0.2203, period length: 2.534; initial concentrations $k_{2b}$=16.0: $A_0$=5.6957, $E_0$=0.4810, and $e_0$=0.2684, period length: 1.583. Other rate constants as in Fig 12.

$$\dot{e} = \underbrace{\frac{k_6 k_{10}}{k_{10} + A}}_{\text{compensated flux } j_6} - k_{11} \cdot e \tag{19}$$

$$\dot{E} = k_{11} \cdot e - \frac{k_7 \cdot E}{k_8 + E} \tag{20}$$

Due to zero-order removal of $E$ ($k_8 \ll E$) the average flux $<j_6> = <k_6 k_{10}/(k_{10}+A)>$ is under homeostatic control. This is seen by setting Eqs 19 and 20 to zero and eliminating the term $k_{11} \cdot e$. This leads to:

$$\frac{k_6 k_{10}}{k_{10} + A} = \frac{k_7 \cdot E}{k_8 + E} \approx k_7 \quad \Rightarrow \quad <\frac{1}{k_{10} + A}> = \frac{k_7}{k_6 k_{10}} \tag{21}$$

Thus, instead of controlling $<A>$ the feedback in Fig 16 controls the property $<1/(k_{10}+A)>$, which is proportional to the average flux $<j_6>$.

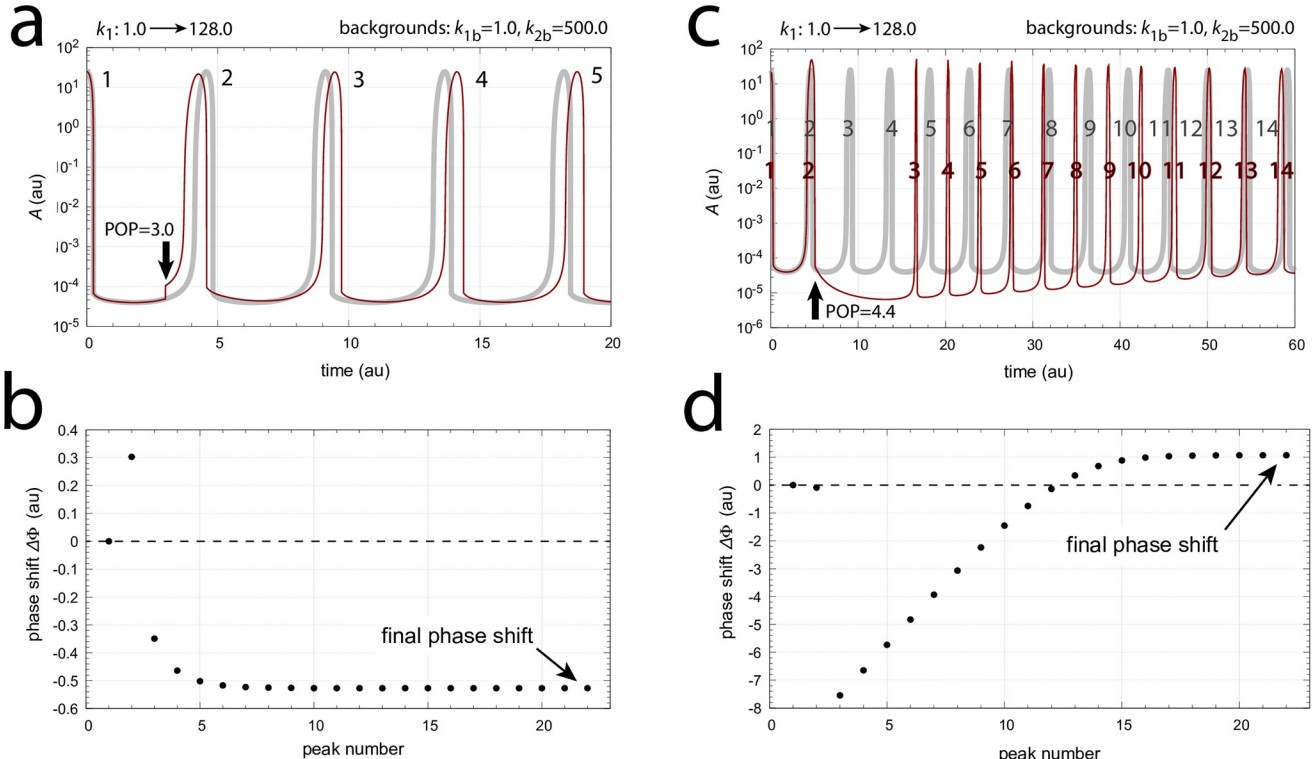

**Fig 14. Two examples of the temporal behavior of phase shifts.** Panel a: The applied phase of perturbation (POP) of a $k_1 : 1.0 \rightarrow 128.0$ step is 3.0 and indicated by the vertical arrow. The gray trace shows the unperturbed oscillation while the red trace shows the effect of the $k_1$ step for the first five peaks. Numbers indicate the perturbed and unperturbed peaks. Panel b: Phase shift $\Delta\Phi$ (Eq 5) as a function of peak number. In total 22 peaks were recorded. Panel c: As panel a, but POP = 4.4 as indicated by the vertical arrow. The picture shows the first fourteen peaks. Panel d: Phase shift $\Delta\Phi$ (Eq 5) as a function of peak number. Although phase shifts are in the beginning negative and reflect delays, the final phase shift is a phase advance. Rate constants: $k_{1b}$=1.0, $k_2$=1.0, $k_{2b}$=500.0, $k_3$=1×10$^4$, $k_4$=1.0, $k_5$=0.1, $k_6$=2.0, $k_7$=$k_8$=1×10$^{-6}$, $k_9$=2.0, $k_{11}$=10.0, $k_{12}$=50.0, $k_{13}$=1×10$^{-6}$, $k_{14}$=50.0, $k_{15}$=10.0, $k_{16}$=$k_{17}$=1×10$^{-6}$, $k_g$=$k_{g3}$=1.0. Initial concentrations: $A_0$=24.921, $E_0$=2.433, $e_0$=3.984, $I_{1,0}$=1.423×10$^4$, $I_{2,0}$=1.153×10$^4$.

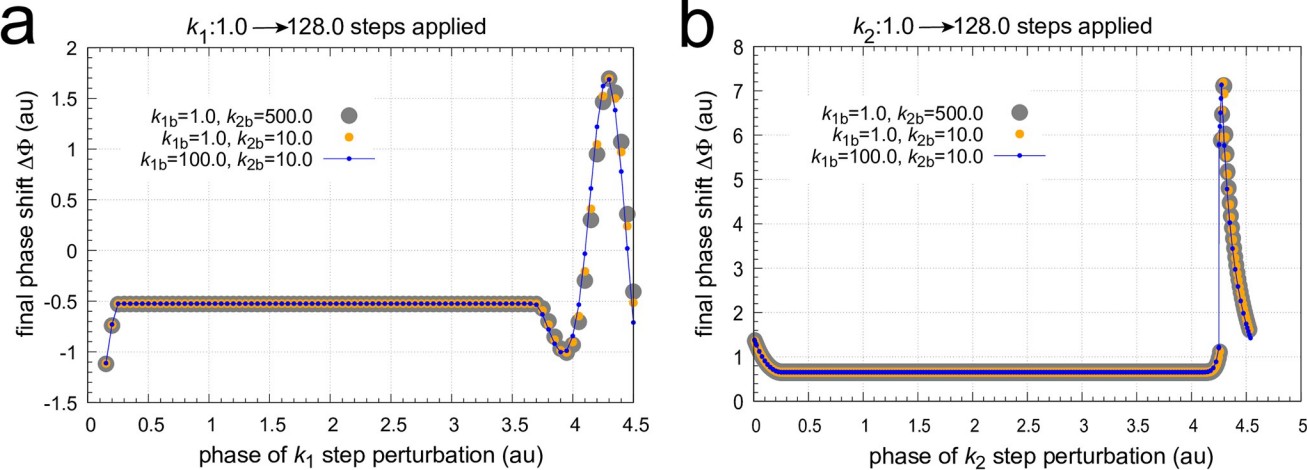

**Fig 15. Phase response curves of the frequency compensated coherent feedback scheme in Fig 4 when $k_1$ and $k_2$ step perturbations are applied at three different backgrounds.** Panel a: The phase response curve for $k_1 : 1.0 \rightarrow 128.0$ steps. Panel b: The phase response curve for $k_2 : 1.0 \rightarrow 128.0$ steps. Initial concentrations for the reference oscillations ($k_1$=$k_2$=1.0): (i) $k_{1b}$=1.0, $k_{2b}$=500.0: see caption Fig 14; (ii) $k_{1b}$=1.0, $k_{2b}$=10.0: $A$=24.839, $E$=2.447, $e$=4.000, $I_1$=3.414×10$^4$, $I_2$=3.375×10$^4$; (iii) $k_{1b}$=100.0, $k_{2b}$=10.0: $A$=24.839, $E$=2.442, $e$=3.987, $I_1$=3.419×10$^4$, $I_2$=3.370×10$^4$. Other rate constants as in Fig 14.

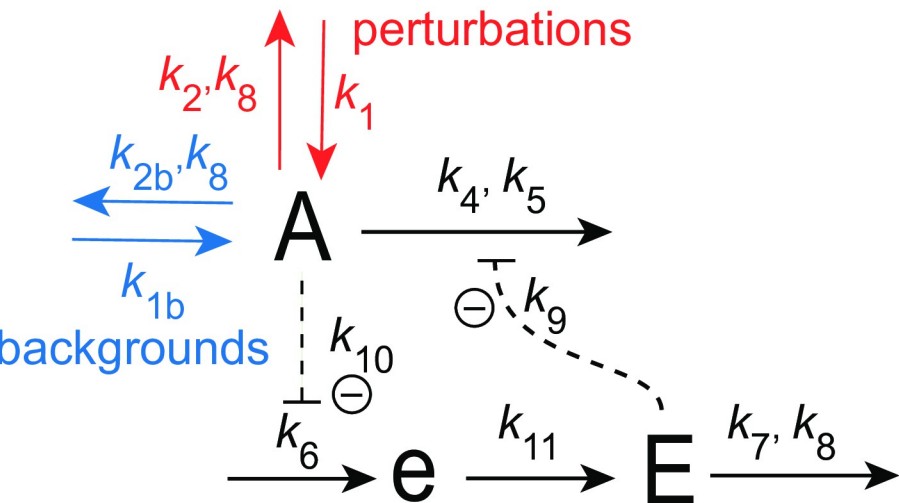

**Fig 16. Motif 8 based negative feedback loop with intermediate *e*.** Reactions outlined in red undergo perturbative steps or pulses, while reactions indicated in blue represent constant backgrounds.

Apart from the condition by Ang et al. [31] referred above, the controller has two operational limits:

(i) a capacity limit to compensate inflows to $A$

and

(ii) the controller's inability to compensate dominating outflows from $A$.

When $k_2 = k_{2b} = 0$, the maximum allowable inflow to $A$ is $k_1 + k_{1b}$, which is balanced by the maximum possible compensatory flux $j_4 = k_4$ when $E \to 0$, i.e. when $j_4 \to k_4$. As the total outflow $k_2 + k_{2b}$ increases the total inflow to $A$, $k_1 + k_{1b}$, can increase accordingly. However, an outflow controller is not able to compensate outflow perturbations, which implies that controller breakdown will occur whenever $k_2 + k_{2b} \geq k_1 + k_{1b}$.

These two scenarios of controller breakdown are presented in Fig 17. In Fig 17a the concentrations of $A$ (left panel) and $E$ (right panel) are shown when $k_1$ increases successively during three phases starting with $k_1 = 5 \times 10^2$ (phase 1), to $k_1 = 8 \times 10^3$ (phase 2), and finally in phase 3 to $k_1 = 1.5 \times 10^4$ while $k_2 = k_{1b} = k_{2b} = 0.0$. In phase 3 $k_1$ exceeds the maximum compensatory flux of $k_4 = 1.0 \times 10^4$ and the controller breaks down: concentration $A$ increases rapidly while $E$ decreases. The left panel in Fig 17a shows in addition, outlined in blue, the calculated value of $<1/(k_{10} + A)>$. Its setpoint, $k_7/(k_6 k_{10}) = 0.5$, is indicated by the thick orange line.

Fig 17b shows the controller's breakdown when the total outflow from $A$ becomes dominant. Here we have constant levels of $k_1 = 1.0$, $k_{1b} = 9.0$, and $k_{2b} = 5.0$. As $k_2$ increases from 0.0 (phase 1) to 3.0 (phase 2), and finally in phase 3 to 5.0, the controller breaks down in phase 3 in the attempt to force the compensatory flux $j_4$ to zero by a steady increase (windup) of $E$. The blue line in the left panel of Fig 17b shows the homeostasis in $<1/(k_{10} + A)>$ during phases 1 and 2 and its breakdown in phase 3.

**M8 coherent feedback: Frequency control and background compensation in frequency resetting.** Fig 18 shows the considered coherent feedback scheme with the M8 feedback from Fig 16 in the center. As in Fig 4 we have that $I_1$ and $I_2$ act as regulators to keep $<E>$ under homeostatic control, which makes the frequency of the oscillator robust against perturbations [39].

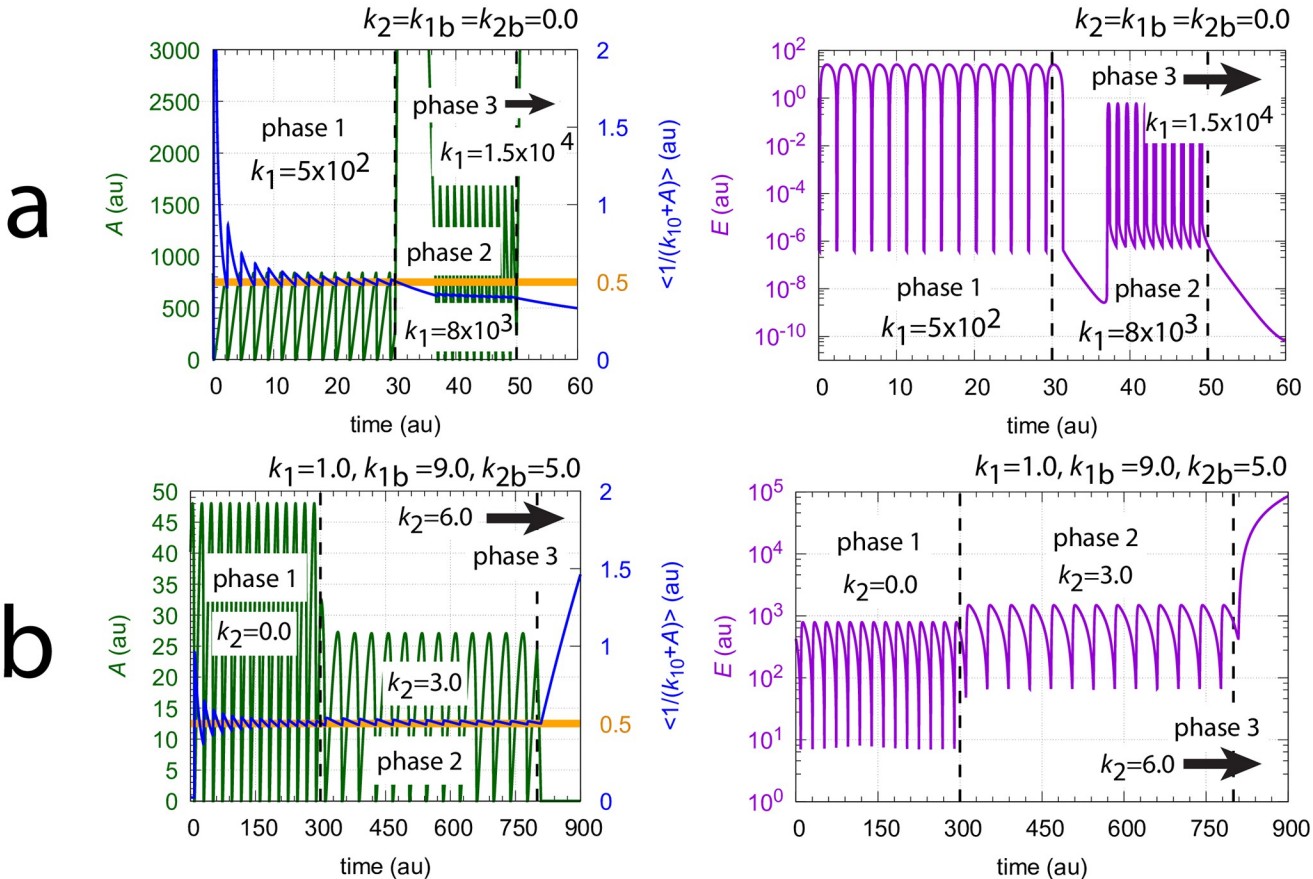

**Fig 17. Breakdown of controller Fig 16 by (a) exceeding the capacity of the compensatory flux $j_4$, and (b) by the dominance of the outflow rate from $A$ with respect to inflows to $A$.** Rate constants, panel a: $k_1$ (phase 1)=$5\times10^2$, $k_1$ (phase 2)=$8\times10^3$, $k_1$ (phase 3)=$1.5\times10^4$, $k_{1b}$=$k_2$=$k_{2b}$=0.0, $k_4$=$1.0\times10^4$, $k_5$=$1.0\times10^{-6}$, $k_6$=$1.0\times10^3$, $k_7$=50.0, $k_8$=$1.0\times10^{-6}$, $k_9$=$k_{10}$=0.1, $k_{11}$=1.0. Initial concentrations, panel a: $A_0$=837.94, $E_0$=1.3246, $e_0$=15.514. Rate constants, panel b: $k_1$=1.0, $k_{1b}$=9.0, $k_2$ (phase 1)=0.0, $k_2$ (phase 2)=3.0, $k_2$ (phase 3)=6.0, $k_{2b}$=5.0. Other rate constants as in figure a. Initial concentrations, panel b: $A_0$=39.942, $E_0$=430.14, $e_0$=2.7212.

The rate equations are:

$$\dot{A} = \underbrace{k_1 - \frac{k_2 \cdot A}{k_{18} + A}}_{\text{perturbations}} + \underbrace{k_{1b} - \frac{k_{2b} \cdot A}{k_{18} + A}}_{\text{backgrounds}} - \left(\frac{k_4 \cdot A}{k_5 + A}\right) \cdot \left(\frac{k_9}{k_9 + E}\right) + k_{g1} \cdot I_1 - \left(\frac{k_{g2} \cdot A}{k_{18} + A}\right) \cdot I_2 \quad (22)$$

$$\dot{e} = \frac{k_6 \cdot k_{10}}{k_{10} + A} - k_{11} \cdot e \quad (23)$$

$$\dot{E} = k_{11} \cdot e - \frac{k_7 \cdot E}{k_8 + E} \quad (24)$$

$$\dot{I}_1 = k_{12} \cdot E - \frac{k_{13} \cdot I_1}{k_{14} + I_1} \quad (25)$$

$$\dot{I}_2 = k_{15} - \left(\frac{k_{16} \cdot I_2}{k_{17} + I_2}\right) \cdot E \quad (26)$$

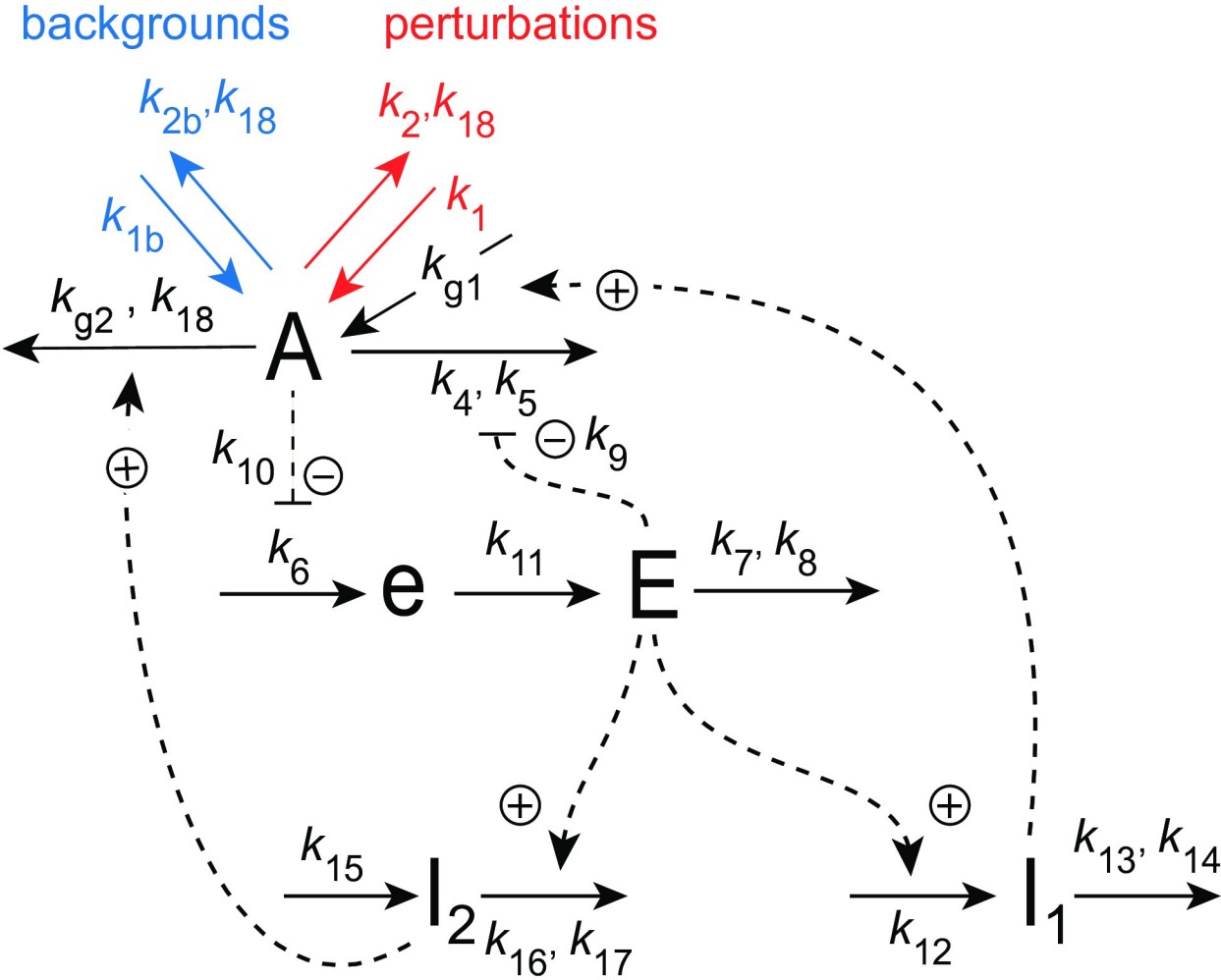

**Fig 18. Coherent feedback scheme using $I_1$ and $I_2$, which control the level of $<j_6>$ and $<E>$ in the central M8-type feedback loop (Fig 16).**

As for the coherent M2 oscillator in Fig 4 we have two setpoints for $<E>$. From the steady state condition of Eqs 25 and 26 together with the zero-order removals of $I_1$ and $I_2$ (i.e. $I_1/(k_{14}+I_1)\approx 1$ and $I_2/(k_{17}+I_2)\approx 1$) we get:

$$<E>_{set}^{I_1} = \frac{k_{13}}{k_{12}}; \qquad <E>_{set}^{I_2} = \frac{k_{15}}{k_{16}} \tag{27}$$

Eliminating $k_{11}\cdot e$ from the steady state expressions of Eqs 23 and 24 leads to the flux control of $j_6 = k_6 k_{10}/(k_{10}+A)$ by

$$<j_6> = k_7 \;\; \text{or} \;\; \left\langle \frac{1}{k_{10}+A} \right\rangle = \frac{k_7}{k_6 k_{10}} \tag{28}$$

Fig 19 shows the behavior of the M8 coherent feedback when the same perturbation and background conditions are applied as in Fig 17b. The breakdown of the controller at $k_2$=6.0 as seen in Fig 17b is now avoided, because $I_1$ is able to compensate for the excess outflow from $A$ (panel c). Panel a, right ordinate shows the homeostasis in $<1/(k_{10}+A)>$, while panels b and d

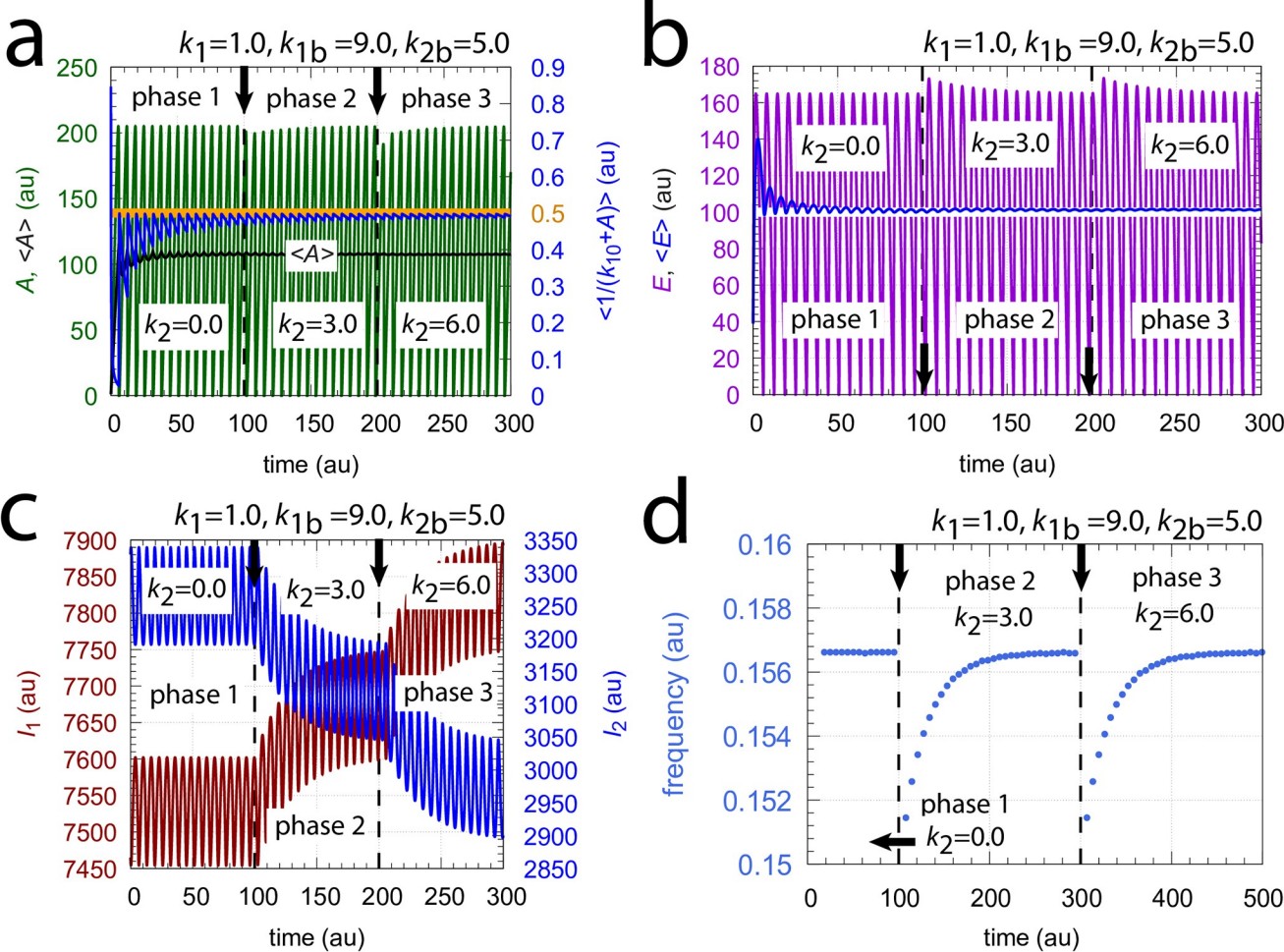

**Fig 19. Homeostasis in frequency, in the average flux $<1/(k_{10}+A)>$, and in the average of $<E>$ by coherent feedback scheme of Fig 18.** At times $t=100$ and $t=200$, indicated by the vertical arrows, $k_2 : 0.0 \to 3.0$ and $k_2 : 3.0 \to 6.0$ steps were applied, respectively with $k_1=1.0$ and backgrounds $k_{1b}=9.0$ & $k_{2b}=5.0$. Panel a, left ordinate: $A$ (outlined in green) and $<A>$ (outlined in black) as functions of time. Panel a, right ordinate: flux $<1/(k_{10}+A)>$ (outlined in blue) as a function of time. Orange line and number indicate the setpoint $<1/(k_{10}+A)>_{set}=k_7/(k_6 k_{10})=0.5$. Panel b: $E$ (outline in purple) and $<E>$ (outlined in blue) as a function of time. The white line indicates the setpoint $<E>_{set}=k_{15}/k_{16}=k_{13}/k_{12}=100.0$. Panel c: $I_1$ and $I_2$ (outlined respectively in red and blue) as a function of time. Panel c: frequency as a function of time. Other rate constants: $k_4=1.0\times10^4$, $k_5=1.0\times10^{-6}$, $k_6=1.0\times10^3$, $k_7=50.0$, $k_8=1.0\times10^{-6}$, $k_9=k_{10}=0.1$, $k_{11}=1.0$, $k_{12}=1.0$, $k_{13}=100.0$, $k_{14}=1.0\times10^{-6}$, $k_{15}=100.0$, $k_{16}=1.0$, $k_{17}=k_{18}=1.0\times10^{-6}$, $k_{g1}=k_{g2}=0.01$. Initial concentrations: $A_0=0.9913$, $E_0=37.985$, $e_0=247.94$, $I_{1,0}=7.466\times10^3$, and $I_{2,0}=3.328\times10^3$. All averages were calculated by Eq 1.

show homeostasis in $<E>$ and the frequency, respectively. Even $<A>$ appears to be under homeostatic control, although there is no explicit mathematical expression for its setpoint!

The coherent M8 feedback scheme shows background compensation when the frequency resetting is tested. Fig 20 shows two resettings, one for $k_1 : 1.0 \to 10.0$ steps (panel a) and the other (panel b) for $k_2 : 0.0 \to 10.0$ steps. For both perturbation types the large orange dots relate to the background combination $k_{1b}=90.0$, $k_{2b}=5.0$ while for the smaller blue dots have the backgrounds $k_{1b}=9.0$, $k_{2b}=50.0$. The panels to the right show the projections of all frequency-time data on to the frequency-time plane. Background compensation is indicated by the complete alignment of the two background combinations.

**Phase response curves of single-loop M8 feedback oscillator.** Next I show how the PRCs of the single-loop feedback in Fig 16 behave as backgrounds change. Figs 21 and 22 show two sets of calculations where $k_{1b}$ and $k_{2b}$ are changed, respectively. The final phase shifts are

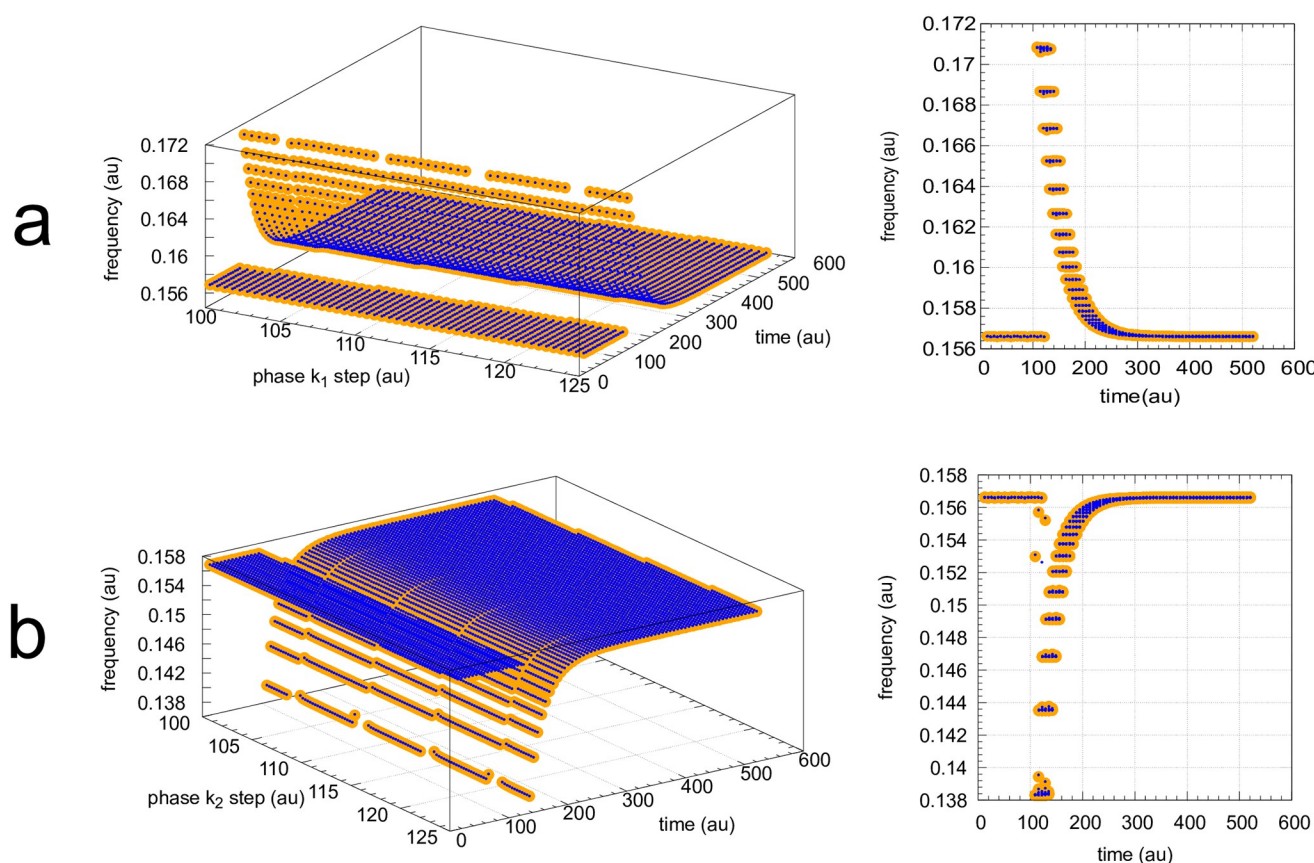

**Fig 20. Background compensation in the frequency resetting of the M8-type oscillator** Fig 18. Figure a, left panel: frequency resetting for $k_1 : 1.0 \rightarrow 10.0$ steps applied at times $t$=100.0 until $t$=125.0 by steps of 0.5 time units. Backgrounds: large orange dots, $k_{1b}$=90.0 & $k_{2b}$=5.0; small blue dots, $k_{1b}$=9.0 & $k_{2b}$=50.0. Figure a, right panel: same data as in left panel, but showing the projections on to the frequency-time plane. Initial concentrations (starting at a maximum of $A$): $A_0$=204.88, $E_0$=20.763, $e_0$=1.563, $I_{1,0}$=3.485×10³, $I_{2,0}$=7.308×10³. $k_2$=1.0, other rate constants as in Fig 19. Figure b, left panel: frequency resetting for $k_2 : 0.0 \rightarrow 10.0$ steps applied at times $t$=100.0 until $t$=125.0 by steps of 0.25 time units. Figure b, right panel: same data as in left panel, but showing the projections on the frequency-time plane. Initial concentrations as in figure a. $k_1$=1.0, other rate constants as in Fig 19.

outlined in orange. Together with the PRC, one cycle of the undisturbed oscillation is shown (outlined in blue) to see the PRC in relationship with the oscillation and the period length. Interestingly, the single-loop M8 oscillator appears to have only positive phase shifts. A direct comparison between the PRCs is made in Fig 23 using a normalized phase of perturbation. The maximum PRC amplitude decreases with increasing $k_{1b}$ background (Fig 23a), which is reminiscent of Weber's law which was observed earlier for this controller (see Fig 11 in [55]). In other words: the response amplitude is diminished at increased backgrounds which are applied in parallel to a constant perturbation to which the controller opposes to. On the other hand, the increase of the PRC amplitude at increased $k_{2b}$'s represents the 'opposite' situation: at increased $k_{2b}$ the controller needs to reduce less $E$ concentrations (or less $<E>$) in order to oppose identical perturbations in $k_1$.

**Coherent M8 feedback oscillator: Background compensation in PRCs.** Finally, I looked at the phase shifts for the $k_1$ and $k_2$ perturbations and background combinations given in Fig 20. An example how final phase shifts have been determined is illustrated in Fig 24. In panel a, a $k_1 : 1.0 \rightarrow 10.0$ step is applied at phase $t = 3.0$ showing the response for the two backgrounds $k_{1b}$=90.0 & $k_{2b}$=5.0 and $k_{1b}$=9.0 & $k_{2b}$=50.0. In panel b a $k_2 : 0.0 \rightarrow 10.0$ step is applied at the

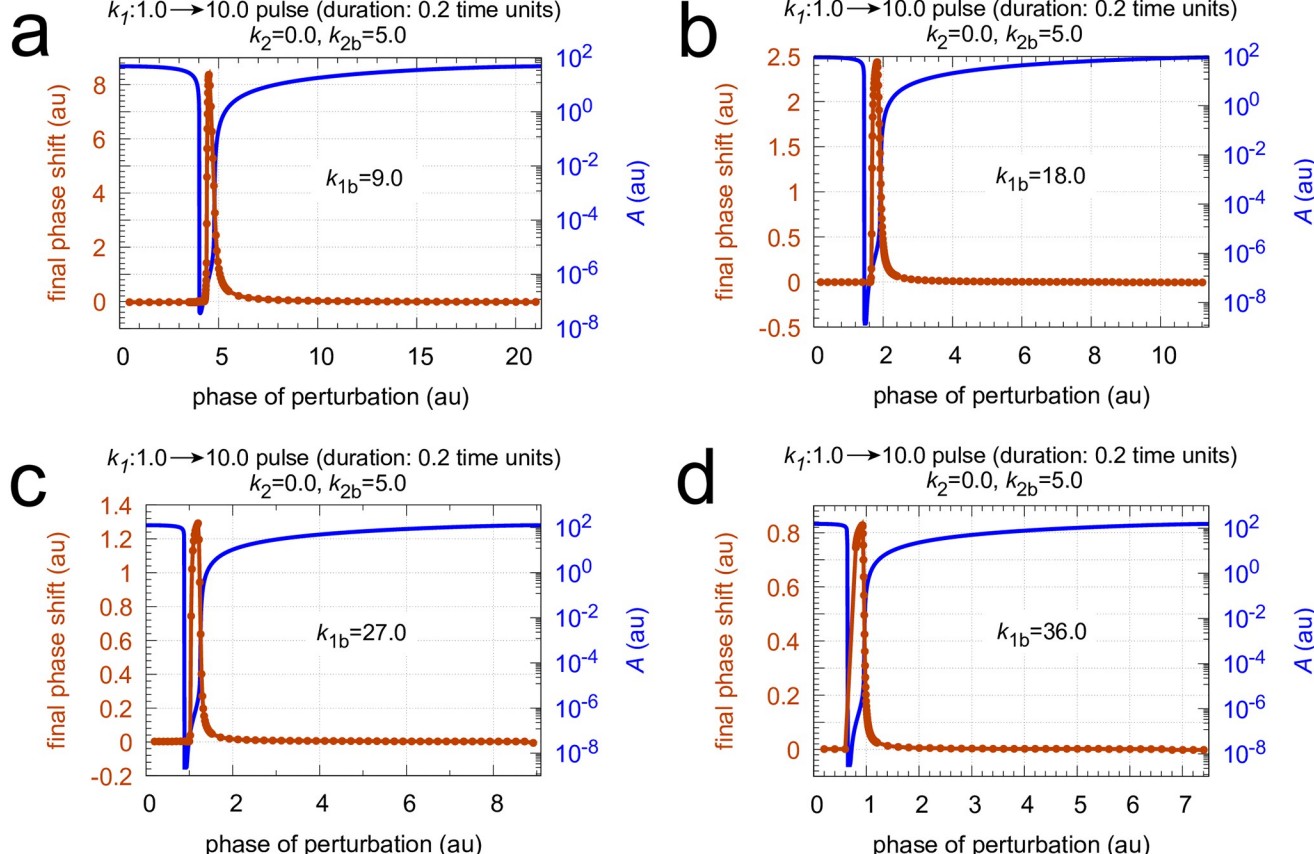

**Fig 21. Phase response curves of oscillator [Fig 16] for $k_1$ pulses 1.0 → 10.0 with a duration of 0.2 time units at different $k_{1b}$ backgrounds.** Panel a: $k_{1b}$=9.0; panel b: $k_{1b}$=18.0; panel c: $k_{1b}$=27.0; panel d: $k_{1b}$=36.0. Note the successive decrease in the maximum phase response amplitude as $k_{1b}$ increases. Rate constants $k_2$=0.0 and $k_{2b}$=5.0. Other rate constants as in [Fig 17]. Initial concentrations: panel a, $A_0$=48.056, $E_0$=199.85, $e_0$=2.1163; unperturbed period = 21.3. Panel b, $A_0$=92.430, $E_0$=71.292, $e_0$=1.1734; unperturbed period = 11.4. Panel c, $A_0$=126.32, $E_0$=43.311, $e_0$=1.0321; unperturbed period = 9.0. Panel d, $A_0$=155.52, $E_0$=31.108, $e_0$=1.1462; unperturbed period = 7.5. All initial concentrations start at an $A$ maximum.

same phase as in panel a testing the same two backgrounds. As for the frequency in [Fig 20] the transient and final phase shifts ΔΦ are independent of the two backgrounds. In fact, the phase response curves ([Fig 25]) show constant final phase shifts which are background compensated.

There is presently no biological evidence for the type of background compensation described here. Although a search in Carl H. Johnson's PRC-Atlas [57] (https://as.vanderbilt.edu/johnsonlab/prcatlas/) resulted in PRCs similar to those of Figs [15] and [25] (see the Atlas PRCs numbered #G/Up-1 [58], #G/Up-2 [59], or PRC #A/Eg-3 [60], respectively) there are no investigations, as far as I can see, where a perturbation background is varied.

For the sake of completeness with respect to PRCs and backgrounds, it should be mentioned that temperature has sometimes been used as a background while applying light or dark pulses as perturbations. An example is the work by Broda et al. (see Fig 4 in [61]) using *Gonyaulax polyhedra* as an organism. The study shows background compensated PRCs with respect to both phase and amplitude when light or dark pulses are applied at 15˚C and 25˚C. On the other hand, studies with other organisms such as Acetabularia [62] and *Neurospora crassa* [63] show PRCs with unaltered phases at different temperature backgrounds but with variable amplitudes. The usage of two different environmental factors such as light and temperature, where one serves as a background and the other as a perturbant act on different

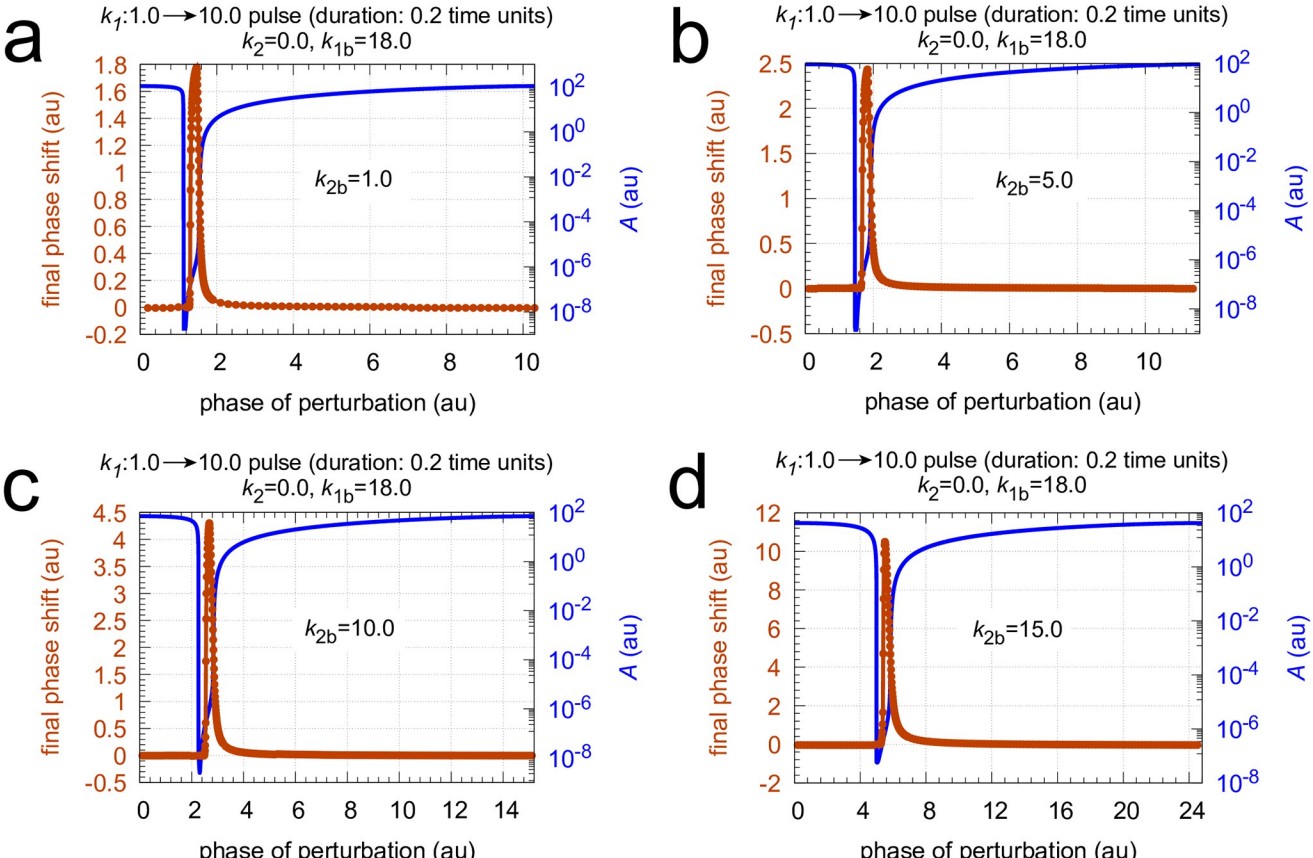

**Fig 22. Phase response curves of oscillator in Fig 16 for $k_1$ pulses 1.0 → 10.0 with a duration of 0.2 time units at different $k_{2b}$ backgrounds.** Panel a: $k_{2b}$=1.0; panel b: $k_{2b}$=5.0; panel c: $k_{2b}$=10.0; panel d: $k_{2b}$=15.0. Note the now successive increase in the maximum phase response amplitude as $k_{2b}$ increases. Rate constants $k_2$=0.0 and $k_{1b}$=18.0. Other rate constants as in Fig 17. Initial concentrations: panel a, $A_0$=108.26, $E_0$=55.385, $e_0$=1.0664; unperturbed period = 10.6. Panel b, $A_0$=92.430, $E_0$=71.322, $e_0$=1.1734; unperturbed period = 11.5. Panel c, $A_0$=70.007, $E_0$=110.01, $e_0$=1.4837; unperturbed period = 15.2. Panel d, $A_0$=41.687, $E_0$=249.90, $e_0$=2.4280; unperturbed period = 24.7. All initial concentrations start at an $A$ maximum.

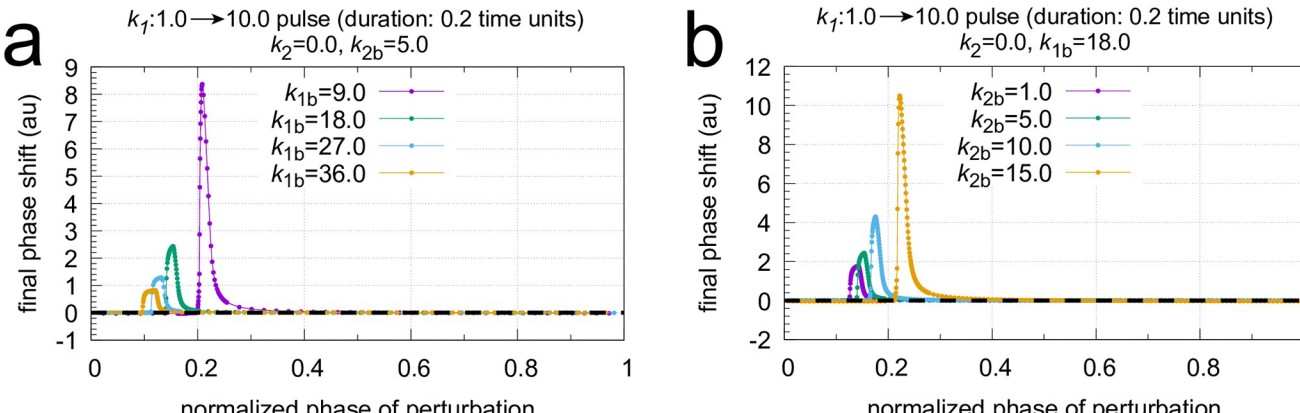

**Fig 23. Influence of $k_{1b}$ and $k_{2b}$ backgrounds on the phase response curves of the oscillator in Fig 16.** To make a direct comparison possible the phases of stimulation are normalized with respect to the oscillator's period length (see Figs 21 and 22). Panel a: comparing phase response curves of Fig 21. Panel b: comparing phase response curves of Fig 22.

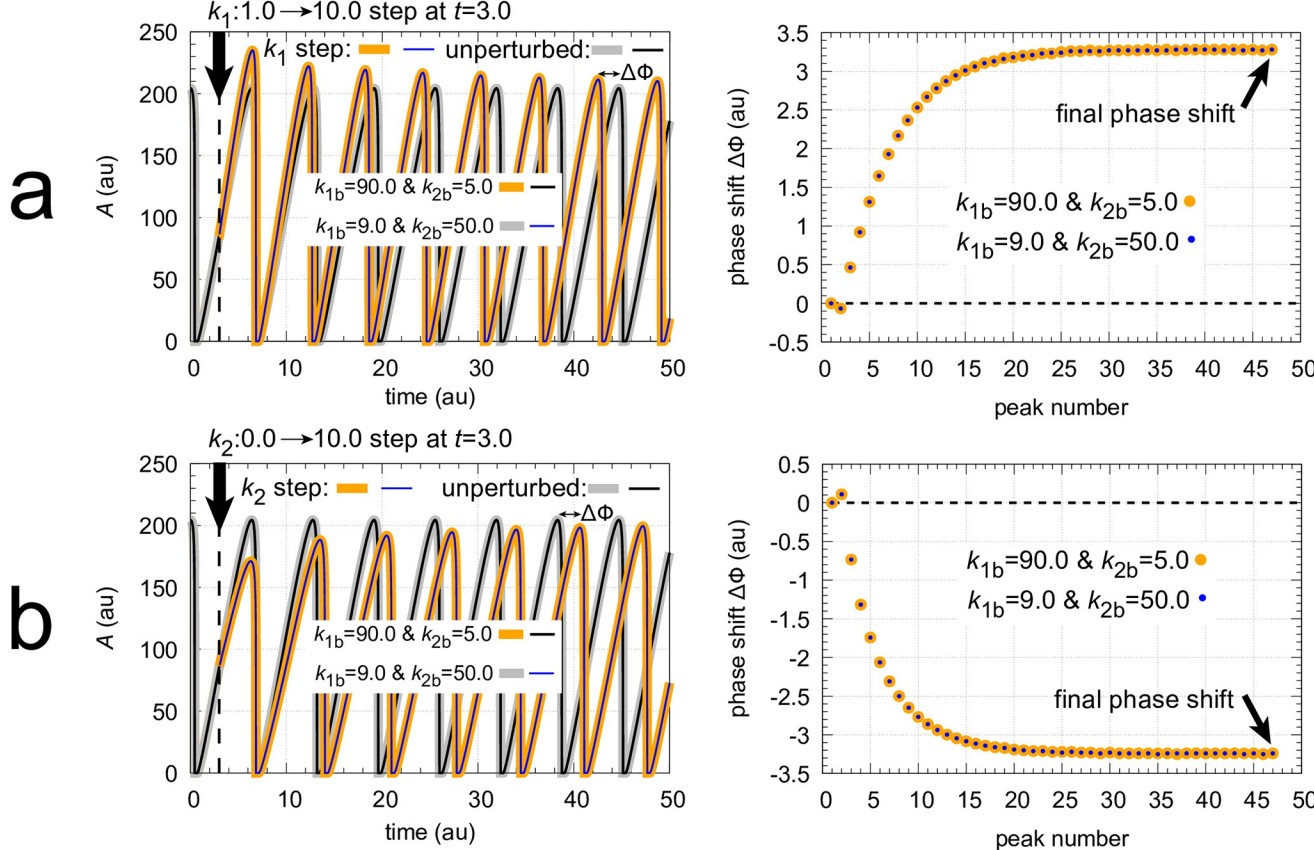

**Fig 24. Background independence of phase shifts in the coherent feedback scheme Fig 18 towards $k_1$ and $k_2$ steps.** Figure a, left panel: Concentration of $A$ at the two background combinations ($k_{1b}$=90.0 & $k_{2b}$=5.0) and ($k_{1b}$=9.0 & $k_{2b}$=50.0) when a $k_1 : 1.0 \rightarrow 10.0$ step indicated by the vertical arrow is applied at $t$=3.0. For the background combination $k_{1b}$=90.0 & $k_{2b}$=5.0 the thin black line shows $A$ of the unperturbed oscillator, while the thick orange line shows the effect of the $k_1$ step. Initial concentrations are: $A_0$=204.34, $E_0$=20.838, $e_0$=1.566, $I_{1,0}$=4.935×10^4, $I_{2,0}$=5.319×10^4. For the background combination $k_{1b}$=9.0 & $k_{2b}$=50.0 the thick gray line shows the unperturbed oscillator while the thin blue line shows the effect of the $k_1$ step. Initial concentrations are: $A_0$=204.36, $E_0$=20.846, $e_0$=1.567, $I_{1,0}$=5.565×10^4, $I_{2,0}$=4.689×10^4. The phase shift $\Delta\Phi$ between unperturbed and perturbed peaks is indicated in the upper right corner of the graph. Rate constant $k_2$=0.0. The right panel of figure a shows that the transient phase shifts shown as a function of peak number are independent of the two backgrounds. Figure b, left panel: Concentration of $A$ for the two background combinations above, but a $k_2 : 0 \rightarrow 10.0$ step is now applied at $t$=3.0. Initial concentrations as for figure a. Rate constant $k_1$=1.0. The right panel of figure b shows that the phase shifts are independent of the two background combinations. Other rate constants as in Fig 19.

reaction channels. This situation is more complex than the one considered here where background and perturbation act on the same controlled variable. The question under what conditions coherent feedback may lead to background compensation when background and perturbation are different, as for example temperature and light, will need further investigations.

## Summary and outlook

I have shown that coherent feedback oscillators have the ability to compensate their frequency resetting and phase shifts against different but constant backgrounds. This indicates that these systems, either oscillatory or nonoscillatory [18] seem to have the potential to 'ignore' ambient backgrounds such as noise. Classical mechanisms to deal with ambient noise can be an increase of the call amplitude, known as the Lombard Effect [64, 65], or a change of the call

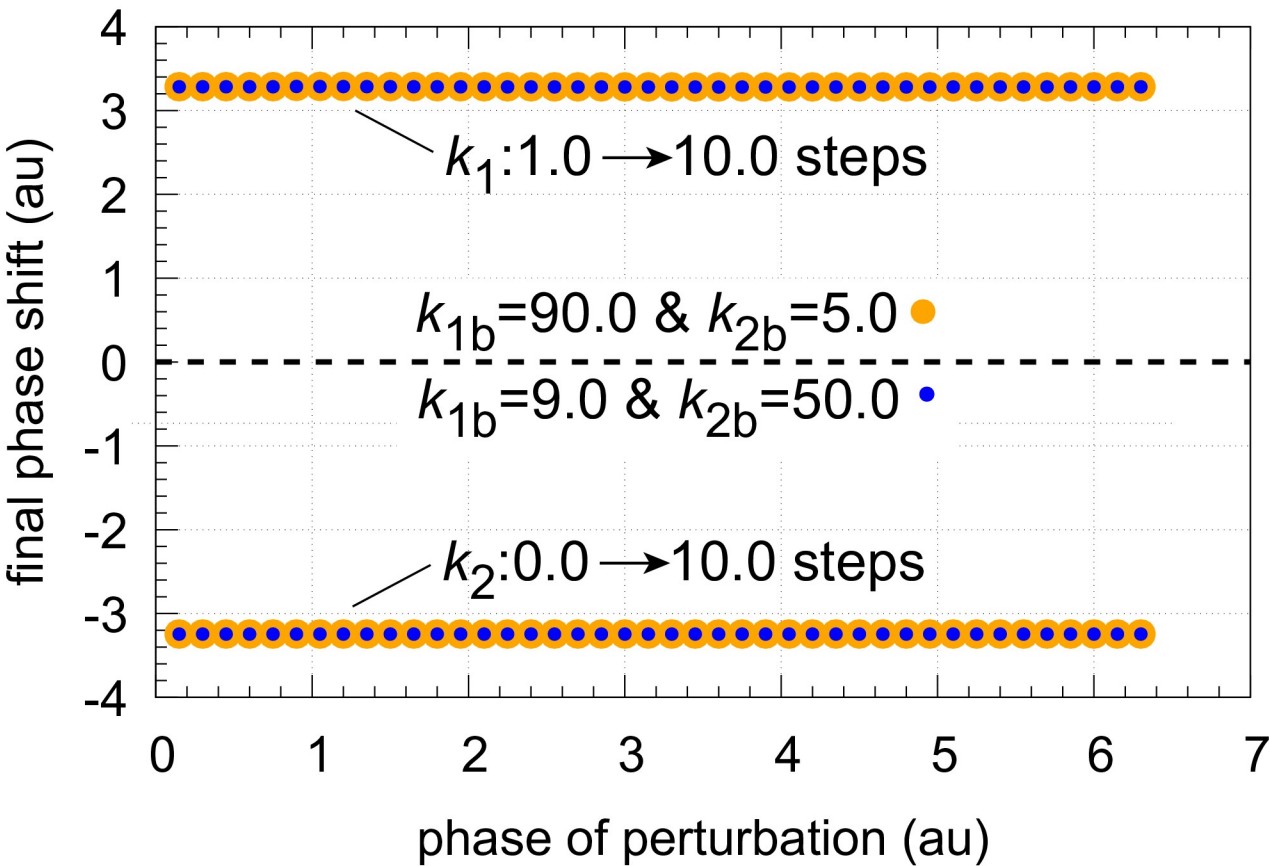

**Fig 25. Final phase shift (see Fig 24, right panels) as a function of phase of perturbation for $k_1 : 1.0 \rightarrow 10.0$ and $k_2 : 0.0 \rightarrow 10.0$ steps at backgrounds $k_{1b} = 90.0$ & $k_{2b} = 5.0$ (orange dots) and $k_{1b} = 9.0$ & $k_{2b} = 50.0$ (blue dots).** Initial concentrations and rate constants as in Fig 24.

frequency [66]. However, the mechanism of background compensation by coherent feedback is different, as it actively compensates for the background by feedback. Concerning the question how the brain processes sound, it has been found that in bats, which live in quite noisy environments, the auditory cortex can actively adjust and improve auditory signal processing by frequency-specific feedback loops between thalamus and cortex (see chapter *The Cerebral Cortex Modulates Sensory Processing in Subcortical Auditory Areas*, page 670ff, in Ref [67]. To what extent background compensation may be used, for example via cortex-thalamus or other brain feedbacks, to recognize specific call patterns within a noisy environment is not known, but it appears interesting to do further research in this direction.

## Supporting information

**S1 Program. Python scripts.** A zip-file with Python and Matlab scripts showing results from Figs 3, 5, 7, 8, 12, 14, 17, 19, and 24.
(ZIP)

**S1 Movie. Animations of Fig 9.** The Quicktime movie files show the trajectories in phase space when $k_1$ and $k_2$ step perturbations are applied. The moving cursor has a length of 1 time unit. The movies also show the preservations of the limit cycles after the step perturbations

when limit cycles are projected on to the *A-E* phase space.
(ZIP)

**S2 Movie. Animation of Fig 10.** The movie shows Fig 10 from various viewing angles.
(ZIP)

**S3 Movie. Animation of Fig 11.** The movie shows Fig 11 from various viewing angles.
(ZIP)

## Author Contributions

**Conceptualization:** Peter Ruoff.

**Formal analysis:** Peter Ruoff.

**Investigation:** Peter Ruoff.

**Methodology:** Peter Ruoff.

**Project administration:** Peter Ruoff.

**Resources:** Peter Ruoff.

**Software:** Peter Ruoff.

**Validation:** Peter Ruoff.

**Visualization:** Peter Ruoff.

**Writing – original draft:** Peter Ruoff.

**Writing – review & editing:** Peter Ruoff.

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
