## [Decision Letter · Decision Letter 0]

19 Jul 2024

PONE-D-24-22584Background compensation revisited: Conserved phase response curves in frequency controlled homeostats with coherent feedbackPLOS ONE

Dear Dr. Ruoff,

Thank you for submitting your manuscript to PLOS ONE. After careful consideration, we feel that it has merit but does not fully meet PLOS ONE’s publication criteria as it currently stands. Therefore, we invite you to submit a revised version of the manuscript that addresses the points raised during the review process.

We look forward to receiving your revised manuscript.

Kind regards,

Jae Kyoung Kim

Academic Editor

PLOS ONE

Reviewers' comments:

Reviewer's Responses to Questions

**Comments to the Author**

1. Is the manuscript technically sound, and do the data support the conclusions?

Reviewer #1: Yes

Reviewer #2: Yes

2. Has the statistical analysis been performed appropriately and rigorously? 

Reviewer #1: Yes

Reviewer #2: N/A

3. Have the authors made all data underlying the findings in their manuscript fully available?

Reviewer #1: Yes

Reviewer #2: Yes

4. Is the manuscript presented in an intelligible fashion and written in standard English?

Reviewer #1: Yes

Reviewer #2: Yes

5. Review Comments to the Author

Reviewer #1: This is an interesting publication, and a strong contribution to the ongoing effort to understand biological control systems, in this case the responses of regulated biological oscillators. The work is carried out to a high technical standard and described clearly and completely.

I have only two small suggestions, one about content and one about style:

1) There is a great deal of material covered, and the sheer number of figures and results could make it difficult for a reader to follow their way through the paper. I wonder if it would be possible to provide a preview/outline at the start of the Results section, to give the reader a sense of what is about to be covered, and where to seek out results of particular types. Something like a bulleted list of the types of perturbations and responses to be investigated, pointing out where each will be addressed? It could make navigating the manuscript less daunting, especially for a reader who might arrive looking for a specific type of controller/response's results. (Alternatively, perhaps a summary/overview at the end of the Results section, collecting and summarizing the results that have been presented.)

2) The main body of the text uses the first person pronoun, which makes perfect sense for a single-author paper. But the Abstract uses "we"; should that also be "I"? This is an actual question rather than a recommendation: I'm really not sure which is more correct, in an Abstract. It may be a question for the editors rather than the author.

Reviewer #2: An interesting manuscript. The results are valuable and are clearly articulated. I would recommend expanding the motivation and biological context to strengthen the case that this work is directly related to behaviours of regulated biomolecular oscillators. I also recommend streamlining the presentation by removing some of the figures (or moving them to the supplement). I’ve made specific recommendations below.

P3: It would be worthwhile to provide some review/context for motif indices from ref 31.?

P5: The justification for approx. 1 in eqn 9 is not clear (i.e. why k7 << Ess is guaranteed.)

P7-8 figs 7-8-9. I had trouble following the argument that ‘While Fig 9 provides insights into why the resetting of k1 steps is markedly 195 different from those of k2 steps.’ The responses in fig 9 are not qualitatively different. Is the comparison of size of response in 6D and 7D the most relevant feature here? Some additional guidance in interpreting these figures (which are dense) would be useful.

P8: Similarly: “I tested whether background 196 compensation is operative for both k1 and k2 step perturbations. This is shown in Figs 10 and 11.” More guidance is needed to interpret these results.

P9: Fig 12 is perhaps not needed: PRC construction has already been illustrated. Likewise Figures 14, 24 and 25

P10: “Surprisingly, this constant phase shift zone resembles that of a dead zone,”

Can anything more be said about this? Seems a remarkable result.

P12: space needed :’ are changed,respectively.”

6. PLOS authors have the option to publish the peer review history of their article (what does this mean?). If published, this will include your full peer review and any attached files.

Reviewer #1: No

Reviewer #2: No

---

## [Author Response · Author response to Decision Letter 0]

6 Aug 2024

Please, see attached file "Response to Reviewers".

---

## [Decision Letter · Decision Letter 1]

19 Aug 2024

Background compensation revisited: Conserved phase response curves in frequency controlled homeostats with coherent feedback

PONE-D-24-22584R1

Dear Dr. Peter Ruoff,

We’re pleased to inform you that your manuscript has been judged scientifically suitable for publication and will be formally accepted for publication once it meets all outstanding technical requirements.

Kind regards,

Jae Kyoung Kim

Academic Editor

PLOS ONE

Additional Editor Comments (optional):

Reviewers' comments:

Reviewer's Responses to Questions

**Comments to the Author**

1. If the authors have adequately addressed your comments raised in a previous round of review and you feel that this manuscript is now acceptable for publication, you may indicate that here to bypass the “Comments to the Author” section, enter your conflict of interest statement in the “Confidential to Editor” section, and submit your "Accept" recommendation.

Reviewer #1: All comments have been addressed

Reviewer #2: (No Response)

2. Is the manuscript technically sound, and do the data support the conclusions?

Reviewer #1: Yes

Reviewer #2: (No Response)

3. Has the statistical analysis been performed appropriately and rigorously? 

Reviewer #1: Yes

Reviewer #2: (No Response)

4. Have the authors made all data underlying the findings in their manuscript fully available?

Reviewer #1: Yes

Reviewer #2: (No Response)

5. Is the manuscript presented in an intelligible fashion and written in standard English?

Reviewer #1: Yes

Reviewer #2: (No Response)

6. Review Comments to the Author

Reviewer #1: My comments have been addressed.

There is apparently a minimum character count in this box, so I will continue to type things until the system agrees to allow me to submit.

Reviewer #2: (No Response)

7. PLOS authors have the option to publish the peer review history of their article (what does this mean?). If published, this will include your full peer review and any attached files.

Reviewer #1: No

Reviewer #2: No

---

## [Editor Report · Acceptance letter]

23 Aug 2024

PONE-D-24-22584R1 

PLOS ONE

Dear Dr. Ruoff, 

I'm pleased to inform you that your manuscript has been deemed suitable for publication in PLOS ONE. Congratulations! Your manuscript is now being handed over to our production team.

Kind regards, 

on behalf of

Prof Jae Kyoung Kim 

Academic Editor

PLOS ONE